# Imprints of increases in evapotranspiration on decreases in streamflow during dry periods, a large-sample analysis in Germany

Giulia Bruno[1,2,3,4], Laurent Strohmenger[1], Doris Duethmann[1,5]

[1]Department of Ecohydrology and Biogeochemistry, Leibniz Institute of Freshwater Ecology and Inland Fisheries (IGB), Berlin, 12587, Germany
[2] now at: Swiss Federal Institute for Snow and Avalanche Research SLF, Davos Dorf, 7260, Switzerland
[3] now at: Institute for Atmospheric and Climate Science, ETH Zürich, Zürich, 8092, Switzerland
[4] now at: Climate Change, Extremes and Natural Hazards in Alpine Regions Research Center CERC, Davos Dorf, 7260, Switzerland
[5] now at: State Office for the Environment Rhineland-Palatinate, Mainz, 55116, Germany

*Correspondence to*: Giulia Bruno (giulia.bruno@slf.ch)

**Abstract.** Decreases in streamflow ($Q$) during dry periods can negatively affect river ecosystems and human societies, and understanding their causes is crucial to anticipate them. The contribution of increases in catchment actual evapotranspiration ($E$) to decreases in $Q$ during dry periods remains poorly quantified. To address this gap, we performed a data-based analysis for 363 small ($< 1000$ km$^2$) catchments without substantial water management influences in Germany over 1970–2019. We quantified trends in the magnitude of summer low flows, i.e., the minimum 7-day $Q$ during summer months ($7dQ_{min, JJA}$). We attributed these trends to their main potential predictors, namely, long-term variations in $E$, summer precipitation, $P$, as well as spring and winter $P$ as proxies for storage. Furthermore, we assessed potential changes in the annual $P$-$Q$ relationship of the catchments during a multi-year drought in the early 1990s, and investigated whether these changes were related with trends and anomalies in $E$ and $P$. Summer low flows generally showed a decreasing tendency (median trend of -3.7 % decade$^{-1}$ and interquartile range of -7.5/-0.6 % decade$^{-1}$ across all catchments), significant negative trends in 31 % of the catchments, and significant positive trends in 2 % of them only. Increases in $E$ were a relevant driver of these decreases particularly in relatively more arid eastern catchments (contribution to long-term dynamics of $7dQ_{min, JJA}$ of 35 % based on multiple linear regression, and correlation coefficient between trends in $7dQ_{min, JJA}$ and in $E$ of -0.74). Changes in the $P$-$Q$ relationship occurred in 26 % of the catchments that experienced a multi-year drought between 1989 and 1993, with lower $Q$ than expected from the relationship before the drought. These changes occurred in catchments with concurrent strong increases in $E$ (median trend of 6.1 % decade$^{-1}$). Our findings point to the importance of increases in $E$, especially in more arid catchments, when assessing potential future decreases in $Q$ during dry periods for water management and climate adaptation strategies.

## 1 Introduction

Long-term decreases in low flows and alterations in streamflow ($Q$) generation under extended dryness can alter river habitat availability (Hauer et al., 2013; John et al., 2022) and human water supply (Garreaud et al., 2017; Montanari et al., 2023). As an example, Montanari et al. (2023) showed that the magnitude of summer low flows in the Po River (northern Italy) has

declined since the 1940s, with the minimum in 2022. The 2022 extreme event had severe ecological and socio-economic impacts, through deterioration of water quality, reductions in power production (Montanari et al., 2023 and references therein), and restrictions on public water use (Avanzi et al., 2024) for instance.

The magnitude of low flows (or shortly, low flows in the following) has decreased in various regions worldwide over recent decades (Stahl et al., 2010; Fangmann et al., 2013; Thomas et al., 2015; Bormann & Pinter, 2017; Hammond et al., 2022; Chagas et al., 2022). Stahl et al. (2010) analysed summer low flows for 441 small catchments (area < 1000 km$^2$) in Europe, and they found mostly decreases in summer low flows, i.e., drying conditions, between 1962 and 2004. Specifically, Stahl et al. (2010) reported decreases (increases) in minimum 7-day $Q$, typically used to assess potential ecological impacts, for 46 (23) % of the catchments and in minimum 30-day $Q$, typically used to assess potential impacts on human societies, for 42 (24) % of the catchments. Bormann & Pinter (2017) later studied long-term variations in annual low flows for 79 small-to-large catchments (481–159300 km$^2$) in Germany between 1950 and 2013, and they reported significant increases, i.e., wetting conditions, in minimum 7-day $Q$ for 26 % of the catchments and decreases for 4 % of them. Bormann & Pinter (2017) showed that increases in low flows occurred in southern and western Germany, while decreases occurred in the north-east of the country. They argued that changes in climate and human activities drove this spatial pattern, with increases in air temperature leading to increases in winter low flows in large, snowmelt-affected catchments in western Germany and changes in human activities, such as mining, to the decreases in the north-east of the country. Small catchments without substantial influences of water management (or shortly, small catchments henceforth) allow the study of changes mainly driven by climate (Thomas et al., 2015; Hodgkins et al., 2024). Furthermore, focusing only on the summer season avoids influences of snow processes into the generation of low flows (Floriancic et al., 2020). Yet, an updated assessment of changes in summer low flows for small catchments in Germany over recent decades is lacking.

Long-term decreases in summer low flows in small catchments may originate from long-term increases in catchment actual evapotranspiration ($E$), decreases in precipitation ($P$) during the summer season, and decreases in water storage in the catchments ($S$, mainly in the soil and groundwater in catchments with little influence of snow, Montanari et al., 2023). Over recent decades, $E$ has increased in many regions following changes in climate and land cover (Duethmann and Blöschl, 2018; Teuling et al., 2019; Yang et al., 2023; Bruno & Duethmann, 2024). These increases in $E$ contributed to decreases in annual $Q$ on the long-term (Teuling et al., 2009a; Fischer et al., 2023; Renner and Hauffe, 2024) and during dry years, especially in mountain catchments (Tran et al., 2023). From reanalysis data, Montanari et al. (2023) showed that increases in $E$ and decreases in snow storage co-occurred with decreases in summer low flows in the Po River. The relative importance of long-term variations in different hydro-climatic predictors, including $E$, to decreases in summer low flows has not been quantified though. Dry periods with abnormally low $P$ can extend over multiple years (multi-year droughts). In some catchments, multi-year droughts led to changes in $Q$ generation, with lower $Q$ than expected from the typical annual $P$-$Q$ relationship reported in Australia (Saft et al., 2015), California (Avanzi et al., 2020), Chile (Alvarez-Garreton et al., 2021), and Europe (Massari et al., 2022). Massari et al. (2022) investigated potential changes in the $P$-$Q$ relationship during various multi-year droughts that occurred in 210 catchments in Europe over 1980–2015 (e.g., the multi-year drought in Central Europe in the early 1990s,

Spinoni et al., 2015). They found that these changes occurred in 33 % of the catchments, with a median decrease in $Q$ across the catchments of -28 % compared to the typical relationship. Potential causes of alterations in $Q$ generation during extended dry periods are increases in $E$ and decreases in $S$, with further influence of the severity of $P$ deficits (Saft et al., 2016). Massari et al. (2022) showed positive $E$ anomalies during the investigated multi-year droughts in Europe, which likely contributed to changes in the $P$-$Q$ relationship. Positive $E$ anomalies during multi-year droughts may be caused by high atmospheric evaporative demand in warm-dry conditions and sustained by a depletion of $S$ (Massari et al., 2022). Underlying long-term increases in $E$ may also lead to positive $E$ anomalies during multi-year droughts and thus, to changes in the $P$-$Q$ relationship. Gardiya Weligamage et al. (2023) investigated potential long-term increases in $E$ for six catchments in south-eastern Australia which experienced changes in the $P$-$Q$ relationship during the Millennium drought (ca. 1997–2010). Yet, they showed stable $E$ before and after the drought, and pointed to decreases in $S$ as dominant driver of changes in the $P$-$Q$ relationship during this event. Bruno & Duethmann (2024) reported robust increases in water balance-derived $E$ in small catchments in Germany between the 1970s and 2000s, regardless of uncertainties in $P$ data and in the consideration of potential $S$ changes. However, whether changes in the $P$-$Q$ relationship during the multi-year drought in Germany in the early 1990s occurred in catchments with long-term increases in $E$ has not been assessed yet.

To address the three research gaps delineated before, we set three working hypotheses:

1. Summer low flows decreased in small catchments in Germany between 1970 and 2019.
2. Increases in $E$ were a relevant driver of decreases in summer low flows in small catchments in Germany between 1970 and 2019.
3. During the multi-year drought in Germany in the early 1990s, the occurrence of changes in the annual $P$-$Q$ relationship was related with increases in $E$ over past decades.

To test these working hypotheses, we performed three main analyses (Fig. 1). Firstly, we detected trends in the magnitude of summer low flows (Sect. 2.4). Secondly, we attributed the trends in summer low flows to their main potential predictors (Sect. 2.5). For this trend attribution, we quantified (i) the contribution of each predictor to the long-term temporal dynamics of summer low flows, and (ii) the strength of the spatial coherence between trends in summer low flows and in the considered predictors. Thirdly, we identified catchments that experienced a multi-year drought in the early 1990s, detected changes in their $P$-$Q$ relationship during this event compared to the relationship before the drought, and analysed the underlying trends and anomalies in $E$ and $P$, for catchments with change and no change (Sect. 2.6).

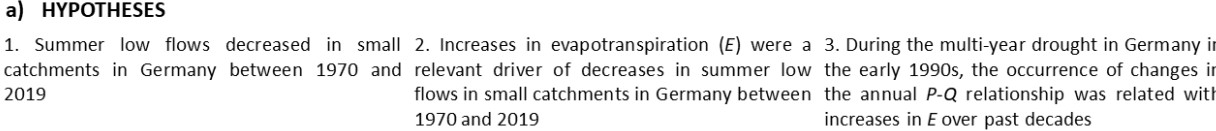

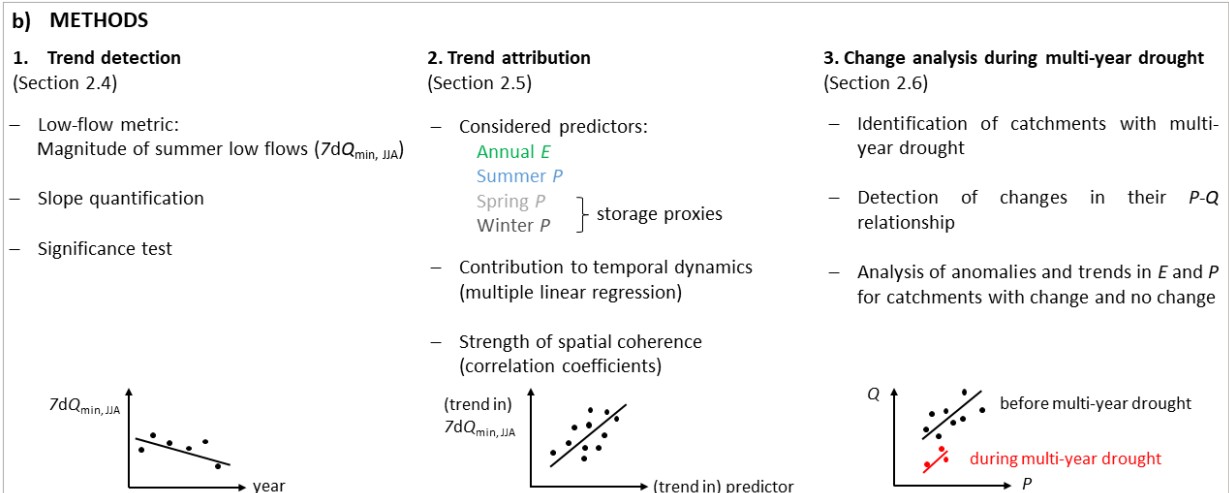

Fig. 1: Study design. (a) Working hypotheses and (b) overview of the methods used to test them.

## 2 Data and methods

### 2.1 Study catchments: streamflow data, clustering, and attributes

We collected daily $Q$ observations and catchment boundaries for 363 catchments in Germany (Fig. 2a) from the Environment Agencies of the German Federal States, the Global Runoff Data Center, and the Global Streamflow Indices and Metadata Archive (Do et al., 2018a; 2018b). These catchments have no substantial influences of water management, according to information from the Environment Agencies, and have an area between 50 and 1000 km$^2$. Furthermore, the selected catchments have $Q$ data covering at least 48 years between 1970 and 2019, with less than 5 % of missing values in each year to ensure sufficient data consistency, and they have mean annual runoff ratio less than 1, to minimize potential issues in annual $Q$ and/or $P$ data. Throughout the manuscript, we defined years on a hydrological basis, starting in November.

To identify regions with homogeneous long-term variations in $Q$ over the study period and ease the trend attribution (Fig. 1b), we grouped the catchments into clusters. To this end, we focused on variations in standardized monthly $Q$ anomalies, rather than variations in a specific low-flow metric, since various metrics can characterize low flows. For comparability across catchments with potentially different hydrological regimes, we computed standardized monthly $Q$ anomalies (or z-scores, $Z_Q$) as:

$$Z_{Q_j}(t) = \frac{Q_j(t) - \overline{Q_J}}{\sigma_{Q_j}} \tag{1}$$

where $\overline{Q_J}$ is the mean and $\sigma_{Q_j}$ the standard deviation in $Q$ for month $j$ ($j = 1…12$) over the study period. We then used a

hierarchical clustering algorithm with Ward's criterion to minimize the variance of $Z_Q$ within the clusters (Ward, 1963). We

set the number of clusters to four (Fig. 2), since the within-cluster variance reduced comparatively less for higher numbers

(Fig. S1).

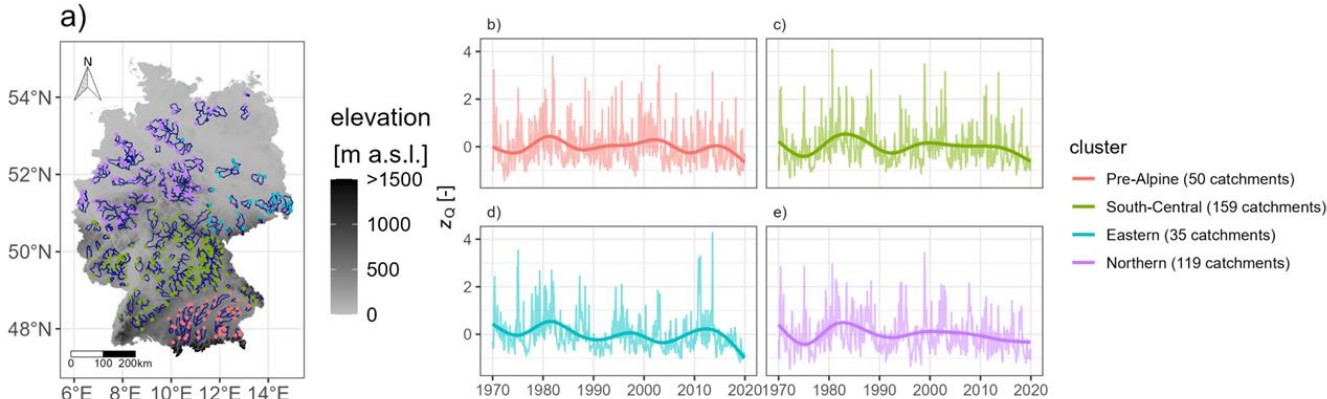

**Fig. 2: Study catchments and streamflow ($Q$) data. (a) Map of catchment boundaries (blue lines) and their outlets (dots, colored according to the cluster they belong to), with elevation from EU-DEM (2016) as background. (b-e) Average monthly standardized $Q$ anomalies ($Z_Q$) across the catchments in the different clusters (solid lines for data smoothed by generalized additive models).**

For catchment characterization, we derived a set of static attributes regarding topography, climate, land cover, and hydrology

as summarized in Table 1.

**Table 1: Summary of attributes for catchment characterization. Attribute, brief description with relevant references, data sources, median (first/third quartile) value across all catchments, and unit. See Fig. 3 and S2 for distributions of the attributes across the different clusters.**

| Attribute | Description | Data source | Median (first/third quartile) | Unit |
|---|---|---|---|---|
| Area | Catchment area from catchment boundaries | Environment Agencies of the German Federal States and the Global Streamflow Indices and Metadata Archive (Do et al., 2018a; 2018b) | 181 (106/331) | km$^2$ |
| Slope | Catchment-average slope | EU Digital Elevation Model (v1.1., EU-DEM, 2016) | 5 (3/6) | ° |
| Mean elevation | Catchment-average elevation | | 396 (216/507) | m a.s.l. |

| Snow fraction | Ratio between $P$ falling as snow (air temperature $< 0°$) and total $P$ | E-OBS dataset (v26.0e, Cornes et al., 2018) | 0.07 (0.05/0.09) | - |
|---|---|---|---|---|
| Precipitation seasonality | Seasonality index from Woods (2009), 0 stands for uniform $P$ throughout the year, 1 for strong peak in summer season, -1 for strong peak in winter season | | 0.04 (-0.03/0.16) | - |
| Aridity index | Ratio between mean daily potential evapotranspiration (Penman-Monteith formulation, Allen et al., 1998) and $P$ | | 0.7 (0.61/0.81) | - |
| Urban areas | Percentage of urban areas, croplands, pastures, and forests within the catchment | Corine 2000 Land Cover data (v20.1, European Environment Agency, 2020) | 5 (3/7) | % |
| Croplands | | | 43 (26/60) | % |
| Pastures | | | 7 (4/13) | % |
| Forests | | | 33 (21/45) | % |
| Sand | Catchment-average percentage of sand, silt, and clay | SoilGrids250m dataset (Hengl et al., 2017) | 40 (34/45) | % |
| Silt | | | 39 (36/43) | % |
| Clay | | | 20 (17/22) | % |
| Runoff ratio | Ratio between mean daily streamflow ($Q$) and $P$ | Environment Agencies of the German Federal States and the Global Runoff Data Center, E-OBS dataset (v26.0e, Cornes et al., 2018) | 0.34 (0.26/0.43) | - |
| Baseflow index | Ratio between baseflow (Ladson et al., 2013) and $Q$ | | 0.66 (0.59/0.74) | - |

According to these attributes (Table 1), the study catchments are predominantly small (median area of 181 km$^2$ across the catchments), flat (median slope of 5 °), and lowland (median mean elevation of 396 m), especially in the eastern and northern clusters (Fig. 3a). They receive little snow (median snow fraction of 0.07) and $P$ is generally distributed rather evenly throughout the year (slightly skewed to the summer season for the pre-Alpine cluster in particular, Fig. 3b). The catchments are mostly humid, with an aridity index occasionally greater than 1, particularly in the eastern cluster (Fig. 3c). The dominant land cover types are croplands and forests, with generally low presence of pastures and urban areas. The catchments have predominantly sandy and silty soils. Runoff ratios are generally low (median of 0.34) and the contribution of baseflow to $Q$ is mostly moderate (median baseflow index of 0.66, Table 1).

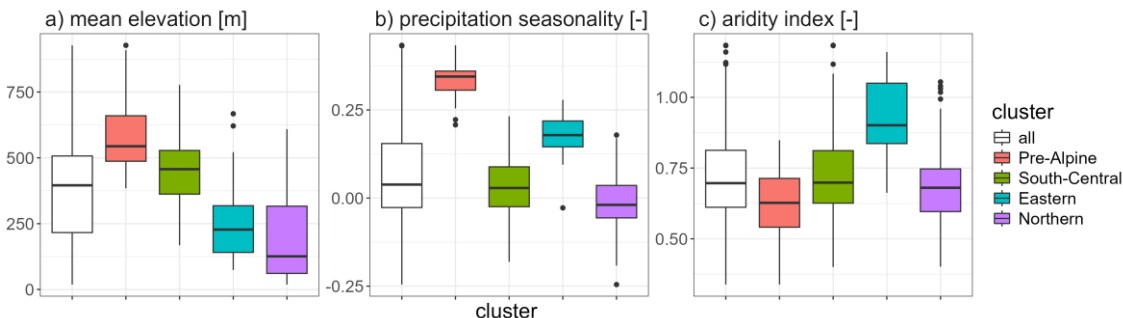

Fig. 3: Selected attributes for the study catchments. Boxplots of (a) mean elevation, (b) precipitation seasonality, and (c) aridity index for all catchments and by cluster. Details on the attributes are in Table 1 and boxplots of additional attributes in Fig. S2.

## 2.2 Precipitation data

Since the focus of our work are long-term variations, we selected a gridded $P$ dataset which interpolates data only from gauges with continuous record over time to minimize inhomogeneities from a time-varying number of gauges. The dataset, previously used in Hoffmann et al. (2018) and Bruno & Duethmann (2024), exploits the SPHEREMAP method (Shephard, 1968; Willmott et al., 1985) for the interpolation and provides daily $P$ fields over Germany at a 0.11° resolution.

We corrected the dataset for gauge undercatch following the method proposed for Germany by Richter (1995):

$$P_{corr} = P_{uncorr} + a P_{uncorr}{}^{b}$$ (2)

with $P_{corr}$ as corrected $P$, $P_{uncorr}$ uncorrected $P$, $a$ and $b$ coefficients which vary with wind exposure of the gauges, precipitation type (rain or snow), and season. Here, we used for all grid cells the coefficients for moderately sheltered locations in Richter (1995), given the low sensitivity of long-term variations in $P$ to the selected coefficients (Duethmann and Blöschl, 2018). To discriminate between rain and snow, we relied on daily air temperature observations from the E-OBS dataset (v26.0e, Cornes et al., 2018). We then computed area-weighted catchment average time series from $P_{corr}$ ($P$ in the following) and $P_{uncorr}$ fields for each catchment. We repeated all analyses (Fig. 1) for both $P$ and $P_{uncorr}$ to verify that the use of the correction procedure does not affect our results.

## 2.3 Water-balance derived catchment evapotranspiration

We derived annual $E$ from the water balance:

$$E = P - Q - \Delta S$$ (3)

with $P$ as annual precipitation, $Q$ as annual streamflow, and $\Delta S$ as annual changes in $S$.

Since long-term data on $S$ across a large number of small catchments are not available, we approximated $S$ with the dynamic storage of the catchments ($S_{dyn}$, that is the portion of $S$ connected to variations in $Q$; Staudinger et al., 2017). Kirchner (2009) showed that $S_{dyn}$ can be derived from the $Q$ time series and recession characteristics of the catchments, under the assumption that $Q$ is mainly controlled by $S_{dyn}$. To verify this assumption for the study catchments, we followed an approach similar to

Kirchner (2009). Specifically, we computed the percentage of winter days over the study period with (i) rising $Q$ ($dQ/dt > 0$) when $P$ exceeds $Q$ and (ii) decreasing $Q$ ($dQ/dt \leq 0$) otherwise. These conditions indicate that other possible storages, that are recharged during $P$ events and then sustain $Q$ during dry periods, are limited in the catchments. Across our catchments, the percentages of days for which $S_{dyn}$ can be thought as the main source of $Q$ spanned between 54 and 72 % meaning that the assumption underlying the method in Kirchner (2009) is met for most of the time in all catchments. We followed the details in Bruno & Duethmann (2024; Text S1 herein) for the derivation of the recession characteristics and $S_{dyn}$. To constrain unrealistic values in $S_{dyn}$, we did not rely on the long-term water balance as in Bruno & Duethmann (2024), but we derived $S_{dyn}$ between $Q1$ ($Q$ exceeded 99% of the days) to avoid numerical issues with low $Q$ and $Q85$ ($Q$ exceeded 15% of the days) to minimize high $Q$ events with relevant surface processes not controlled by $S_{dyn}$ (Staudinger et al., 2017).

## 2.4 Trend detection for summer low flows

We characterized summer low flows in terms of their magnitude, by averaging $Q$ over a moving window of 7 days and then selecting the minimum during the summer months (June, July, and August, $7dQ_{min, JJA}$). To analyse the influence of the adopted low-flow metric on the results, we used two alternative metrics which we derived by extending (i) the moving window to 30 days ($30dQ_{min, JJA}$) and (ii) the summer season from May to October (included, $7dQ_{min, M-O}$). We further considered additional $Q$ metrics, namely annual $Q$ and the centre of mass date of $Q$ ($CMD_Q$). $CMD_Q$ is the time of the year when half of the annual $Q$ has occurred and as such, a metric for $Q$ timing (Court, 1962; Han et al., 2024).

We detected long-term variations through trend analysis. Specifically, we used the Sen's slope estimator (Sen, 1968), as a measure of the magnitude of the trends, and the Mann-Kendall test, with trend-free prewhitening to avoid lag-one autocorrelation, to evaluate their significance (Yue et al., 2002). For all statistical tests throughout the manuscript, we used a 5 % significance level.

## 2.5 Trend attribution

As predictors of trends in summer low flows ($7dQ_{min, JJA}$ in the following), we considered variations in $E$, summer $P$ ($P_{JJA}$) and $S$. We used $P_{MAM}$ and $P_{DJF}$ as proxies of storage recharge in the seasons preceding the summer, following previous works, to overcome the unavailability of long-term data on soil moisture and groundwater storage for the study catchments (Duethmann et al., 2015; Saft et al., 2016; Laaha et al., 2017). To disentangle the relative contribution of these predictors, we performed two analyses.

Firstly, we used multiple linear regression to model the temporal dynamics of summer low flows ($7dQ_{min, JJA}$ in Eq. (4)), from the dynamics of the predictors, both at a catchment- and cluster-scale:

$$7dQ_{min, \, JJA} = \alpha_1 E + \alpha_2 P_{JJA} + \alpha_3 P_{MAM} + \alpha_4 P_{DJF} + \varepsilon \tag{4}$$

with $\alpha_i$ ($i = 1...4$) the regression coefficient for each predictor and $\varepsilon$ the model residuals. We adopted 5-year averages to focus on long-term dynamics and reduce potential uncertainties in water balance-derived $E$. Moreover, for the cluster-scale analysis

we used average time series across the catchments in each cluster to minimize uncertainties in $E$ for specific catchments, while analysing the main signal at a regional scale. We evaluated the considered predictors for multicollinearity and discarded models with variance inflation factor $\geq 10$ (Draper and Smith, 1998). We assessed the performance of the multiple linear regression models in terms of coefficient of determination ($R^2$) between simulated and observed summer low flow. We determined significant predictors according to a two-sided t-test, and expressed the contribution of each predictor $j$ ($j = 1…4$)

as:

$$C_j = \frac{|\tilde{\alpha}_j|}{\sum_{i=1}^{4}|\tilde{\alpha}_i|} * 100 \tag{5}$$

where $\tilde{\alpha}_i$ ($i = 1…4$) are the standardized regression coefficients (i.e., centered around their mean and scaled by their standard deviation) of the considered predictors. We quantified the uncertainty in the regression coefficients as their standard errors. Secondly, we assessed the strength of the spatial coherence between catchment-scale trends in summer low flows and trends in predictors by means of a correlation analysis based on Pearson's correlation coefficient ($r$) and performed for each cluster.

We quantified trends in predictors as described for summer low flows (Sect. 2.4).

We repeated both analyses for the alternative low-flow metrics (Sect. 2.4).

## 2.6 Change analysis during a multi-year drought

We identified catchments that experienced a multi-year drought (or drought in the following) based on annual $P$ standardized anomalies ($Z_P$):

$$Z_P(t) = \frac{P(t) - \bar{P}}{\sigma_P} * 100 \tag{6}$$

where $\bar{P}$ is the mean and $\sigma_P$ the standard deviation in annual $P$ over the study period. We relied on the criteria of Saft et al. (2015) and Massari et al. (2022) to determine start and end years of the drought for each catchment. Firstly, we computed 3-year moving averages of $Z_P$ ($3yZ_P$) to minimize the potential influence of single wet years in the drought identification (Saft et al., 2015). We then considered multi-year droughts as periods with consecutive negative values in $3yZ_P$, by discarding end years with (i) $Z_P > 0.15$ for that year, and (ii) $0 < Z_P < 0.15$ for both that year and the previous one (Saft et al., 2015). Finally,

we retained only catchments that experienced a multi-year drought longer than 3 years with mean $Z_P < -0.8$ and concomitant $Q$ data available (Sect. 2.1), to ensure focusing on relevant events in terms of drought severity (Massari et al., 2022) and data consistency.

For each catchment with a multi-year drought, we identified changes in the $P$-$Q$ relationship during this event compared to the conditions before the drought following the approach of Saft et al. (2015). To ensure that annual $Q$ data is approximately

normally distributed for subsequent parametric analyses, we applied a Box-Cox transformation (Box and Cox, 1964):

$$Q_{BC}(t) = \frac{Q(t)^{\lambda} - 1}{\lambda} \tag{7}$$

where $\lambda$ is a transformation parameter that we determined through maximum likelihood estimation (Massari et al., 2022). We then identified changes in the *P-Q* relationship during the multi-year drought through multiple linear regression (see Sect. 2.5 for methodological details on the regression):

$$Q_{BC} = \beta_1 + \beta_2 I + \beta_3 P + \varepsilon_1 \tag{8}$$

with $Q_{BC}$ as the Box-Cox transformed $Q$ (Eq. (7)) before and during the multi-year drought, $I$ a binary indicator (1 for years in
the multi-year drought and 0 for years before), $P$ annual precipitation before and during the multi-year drought, $\beta_i$ ($i = 1...3$) regression coefficients, and $\varepsilon_1$ model residuals. A significant $\beta_2$ indicates a change in the *P-Q* relationship, as compared to the one before the drought. For catchments where these changes occurred, we quantified their magnitude as:

$$C_{P-Q\ rel} = \frac{(\beta_1 + \beta_2 I + \beta_3 P^* + \varepsilon_1) - (\beta_1 + \beta_3 P^* + \varepsilon_1)}{(\beta_1 + \beta_2 I + \beta_3 P^* + \varepsilon_1)} * 100 \tag{9}$$

with $P^*$ as a representative annual $P$, given by the average between the minimum and average $P$ over the years before and during the multi-year drought (Saft et al., 2015), to account for non-linearities in non-transformed $P$ data.

Finally, for each catchment with a selected multi-year drought, we quantified trends in $E$ and $P$ over the period from 1970 to the end of the multi-year drought (Sect. 2.4), as well as mean annual anomalies (Eq. (6)) in $E$ and $P$ during the drought. We further computed mean annual anomalies in detrended $E$ and $P$. For the detrending, we removed from the original data the linear trend over the years before and during the drought. We then verified whether catchments with change and no change had statistically different distributions in these trends and anomalies according to the Kolmogorov-Smirnov two-sample test.

## 3. Results

### 3.1 Trends in summer low flows

Summer low flows largely decreased in small catchments in Germany between 1970 and 2019 (Fig. 4, S3a-d, Table S1). Mean $7dQ_{min,\ JJA}$ across all catchments showed generally positive anomalies before the 1990s and negative anomalies afterwards (Fig. 4a). Trends in $7dQ_{min,\ JJA}$ were significantly negative in 31 % of the catchments and positive in 2 % of them only (Fig.
4b, Table S1). Across all catchments, trends in summer low flows had a median (interquartile range, IQR) of -3.7 (-7.5/-0.6) % decade$^{-1}$ (Fig. 4c). Focusing on catchments with significant negative trends only, a median decrease of -9.7 % decade$^{-1}$ was observed (Table S1). Each cluster showed negative median trends (the strongest in the eastern cluster equal to -5.8 % decade$^{-1}$, Fig. 4c). Alternative metrics for summer low flows had comparable behaviour, with a median trend across the catchments of -4.2 % decade$^{-1}$ for $30dQ_{min,\ JJA}$ and of -3.1 % decade$^{-1}$ for $7dQ_{min,\ M-O}$, and with significant decreases for 28 % of the
catchments for both metrics (Fig. S4a-f). The other considered $Q$ metrics also decreased over 1970–2019, but less significantly than summer low flows: annual $Q$ showed a median trend of -1.3 % decade$^{-1}$ across all catchments and significant negative trends in 10 % of them, while $CMD_Q$ exhibited a negligible median trend of -0.006 month year$^{-1}$ across all catchments and significant negative trends in 3 % of them (Fig. S4g-l).

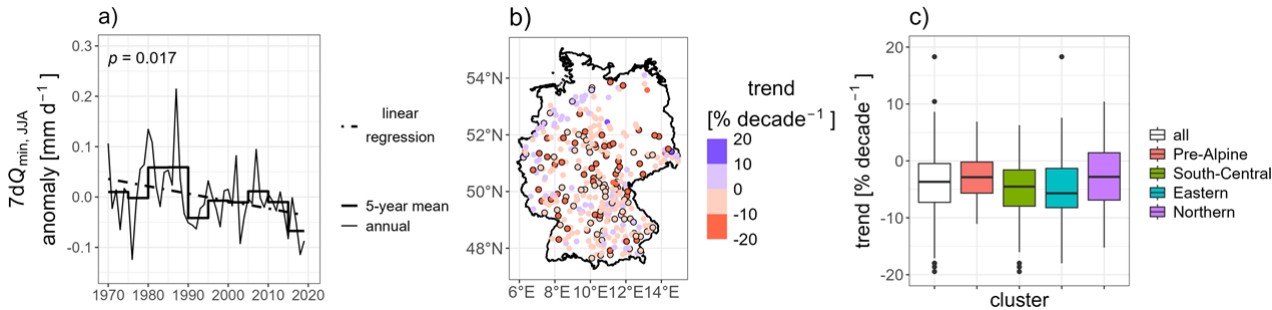

**Fig. 4: Long-term variations in summer low flows (7d$Q_{min, JJA}$) over 1970–2019. (a) Average anomalies across all catchments (average anomalies across the catchments in each cluster in Fig. S6a–d). (b) Map of catchment-scale trends (black edges if significant). (c) Boxplots of trends for all catchments and by cluster.**

## 3.2 Attribution of trends in summer low flows

Annual $E$ increased between 1970 and 2019, with significant increases in 27 % of the catchments and a median trend (IQR) of 1.1 (-0.5/3.2) % decade$^{-1}$ across all of them (Fig. 5a-c). Positive median trends in $E$ occurred in all clusters, and particularly in the northern and eastern ones (3.0 and 2.9 % decade$^{-1}$ respectively, Fig. 5c). Increases in $E$ occurred especially between 1970 and the early 2000s (Fig. 5a), with a median trend (IQR) of 4.2 (1.8/6.9) % decade$^{-1}$ across all catchments if focusing on 1970– 1999 (Fig. S5). Other predictors of trends in summer low flows (i.e., long-term variations in $P_{JJA}$, $P_{MAM}$, and $P_{DJF}$) showed contrasting directions of change over time (Fig. 5d, g, j) and only few significant trends (in < 10 % of the catchments, Fig. 5e, h, k). Trends in $P_{JJA}$ and $P_{MAM}$ further exhibited contrasting signs among the clusters (Fig. 5e, h). $P_{JJA}$ generally decreased in the pre-Alpine cluster (median trend of -1.4 % decade$^{-1}$) and the central one (median trend of -0.8 % decade$^{-1}$), while it generally increased in the eastern (median trend of 2.7 % decade$^{-1}$) and northern clusters (median trend of 1.7 % decade$^{-1}$, Fig. 5f). $P_{MAM}$ underwent general increases in the pre-Alpine cluster (median trend of 1.9 % decade$^{-1}$) and decreases elsewhere (median trends of -0.3 % decade$^{-1}$ in the central cluster, -1.7 % decade$^{-1}$ in the eastern one, and -2.7 % decade$^{-1}$ in the northern cluster, Fig. 5i). Median trends in $P_{DJF}$ were positive for all clusters (Fig. 5l). Similar trends, for both $E$ and seasonal $P$, were obtained when using $P_{uncorr}$ (Fig. S6).

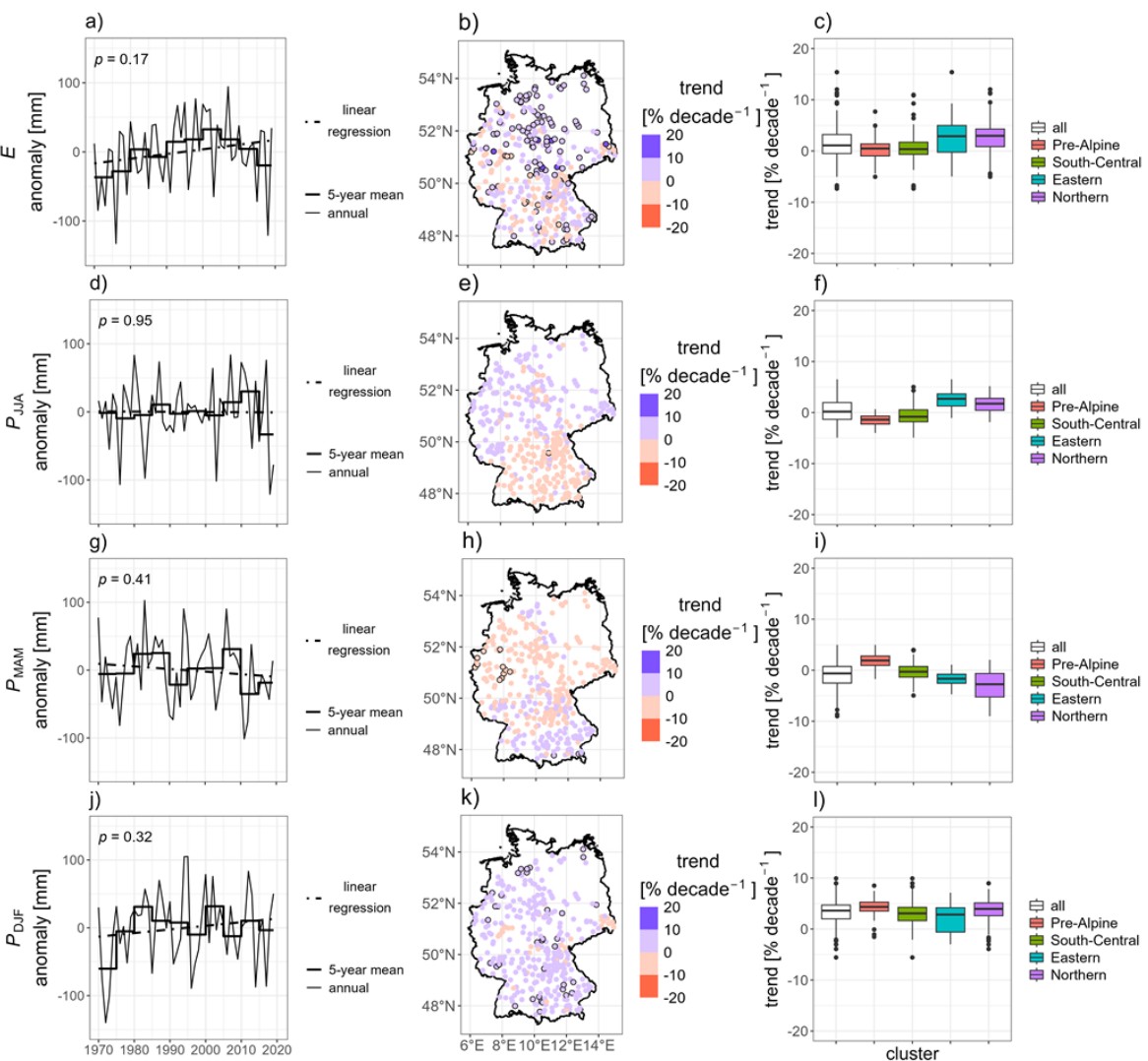

**Fig. 5: Long-term variations in predictors of variations in summer low flows over 1970–2019 (annual evapotranspiration, *E*, panels a–c, precipitation over summer, $P_{JJA}$, panels d–f, spring, $P_{MAM}$, panels g–i, and winter $P_{DJF}$, panels j–l). (a, d, g, and j) Average anomalies across the catchments (average anomalies across the catchments in each cluster in Fig. S3e–t). (b, e, h, and k) Maps of catchment-scale trends (black edges if significant). (c, f, i, and l) Boxplots of trends for all catchments and by cluster.**

Multiple linear regression for predicting long-term dynamics in summer low flows achieved satisfactory performances both at cluster- and catchment-scale (for $7dQ_{min, JJA}$, $R^2 > 0.7$ for each cluster and median $R^2$ of 0.78 across all catchments, Table S2, and Fig. S7a and b). Significant predictors of $7dQ_{min, JJA}$ differed among clusters (Fig. 6, Table S2). In the pre-Alpine cluster, $P_{JJA}$ was the only significant predictor (contribution to the simulation of $7dQ_{min, JJA}$ equal to 46 %). In the south-central cluster, $P_{JJA}$ and $P_{MAM}$ were significant predictors, with comparable contribution to the simulations ($< 40$ % for each predictor). In the eastern and northern clusters, significant predictors were *E*, $P_{JJA}$, and $P_{MAM}$, and they showed similar relative contributions to

the predicted long-term dynamics of 7d$Q_{min, JJA}$(contribution slightly higher for $E$ than for $P_{JJA}$ and $P_{MAM}$ in the eastern cluster, equal to 35 %). Catchment-scale results had similar patterns, despite being more affected by noise than those at the cluster-scale (Fig. S7c and d), as expected. By focusing on significant primary predictors only (i.e., the significant predictors with the highest contribution to the simulations), $P_{JJA}$ was the most recurrent one for catchments in the pre-Alpine cluster, $P_{MAM}$ in the south-central cluster, and $E$ in the eastern and northern clusters (Fig. S7d).

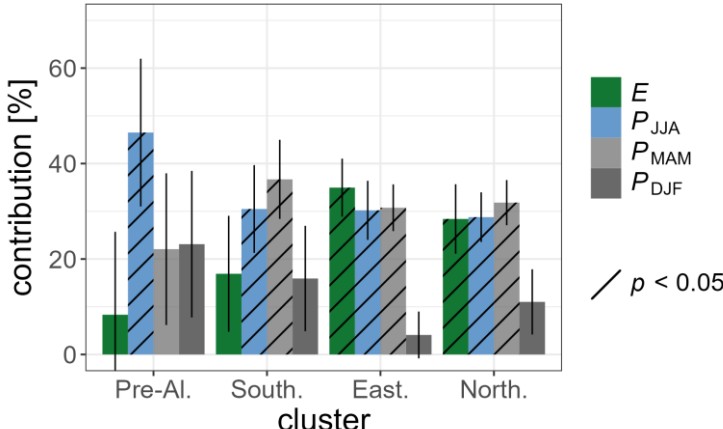

**Fig. 6: Attribution of long-term variations in summer low flows to their predictors (contribution to temporal dynamics): relative contribution of annual evapotranspiration ($E$), summer ($P_{JJA}$), spring ($P_{MAM}$), and winter precipitation ($P_{DJF}$) to the predicted long-term dynamics of summer low flows (7d$Q_{min, JJA}$) from multiple linear regression (Sect. 2.5) for the different clusters. Pre-Al. refers to pre-Alpine, South. to south-central, East. to eastern, and North. to northern cluster. Vertical lines indicate the uncertainty of the regression coefficients.**

The strength of the correlation between catchment-scale trends in summer low flows and in their predictors varied according to the considered predictor and cluster (Fig. 7). By focusing on significant correlations only, trends in 7d$Q_{min, JJA}$ negatively correlated with trends in $E$ in all clusters but the pre-Alpine one ($r$ equal to -0.26 in the south-central, -0.74 in the eastern, and -0.36 in the northern cluster), indicating that decreases in summer low flows corresponded to increases in $E$ and vice versa. Moreover, trends in 7d$Q_{min, JJA}$ positively correlated with trends in $P_{JJA}$ for the pre-Alpine ($r = 0.3$) and northern ($r = 0.25$) clusters, illustrating that decreases in summer low flows paralleled decreases in $P_{JJA}$ and vice versa. Trends in 7d$Q_{min, JJA}$ further showed a positive correlation with trends in $P_{MAM}$ for the pre-Alpine cluster ($r = 0.23$), a negative correlation with trends in $P_{DJF}$ for the eastern cluster ($r = -0.49$), and a positive correlation with trends in $P_{DJF}$ for the northern cluster ($r = 0.16$).

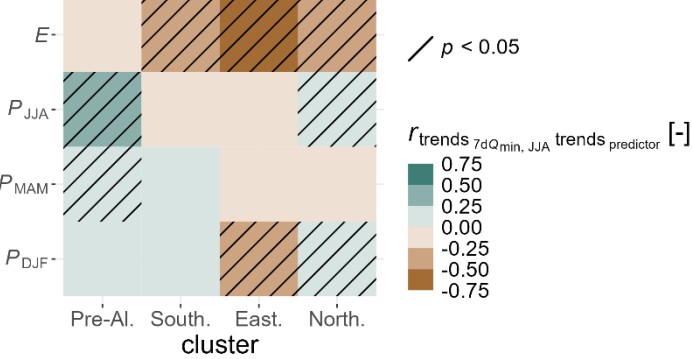

**Fig. 7: Attribution of long-term variations in summer low flows to their predictors (strength of spatial coherence):** Pearson's correlation coefficients ($r$) between catchment-scale trends in summer low flows ($7dQ_{min, JJA}$) and in potential predictors (annual evapotranspiration, $E$, summer precipitation $P_{JJA}$, spring precipitation $P_{MAM}$, and winter precipitation $P_{DJF}$) over 1970–2019, for the catchments in the different clusters. Pre-Al. refers to pre-Alpine, South. to south-central, East. to eastern, and North. to northern cluster.

Correlation between trends in $7dQ_{min, JJA}$ and in their predictors differed also according to the decades considered. Over 1970–1999, trends in $7dQ_{min, JJA}$ for instance negatively correlated with trends in $E$ in all clusters, including the pre-Alpine one ($r = -0.28$, Fig. S8). When using alternative low-flow metrics ($30dQ_{min, JJA}$ and $7dQ_{min, M-O}$) instead of $7dQ_{min, JJA}$ or $P_{uncorr}$ instead of $P$, results were comparable to those presented here for both attribution analyses (not shown).

### 3.3 Changes in the *P-Q* relationship during a multi-year drought and their potential predictors

Fifteen percent of the study catchments experienced a multi-year drought within 1989–1993 (Fig. 8a). Of these, 26 % exhibited a change in the annual *P-Q* relationship during the drought (Fig. 8a). The median magnitude of change across the catchments was of -10 % of *Q* for given *P*, as compared to the relationship before the drought (Fig. 8b). The catchments with change showed largely positive trends in *E* over the period before and during the drought (median of 6.1, IQR of 4.4/9.2 % decade$^{-1}$ across the catchments, Fig. 8c). The catchments with no change had mostly small trends in *E* before and during the drought (median of -0.5, IQR of -0.5/3.4 % decade$^{-1}$, distribution statistically different from the one for catchments with change, Fig. 8c). Trends in annual *P* before and during the drought were generally positive for both catchments with and without change (Fig. 8c). Mean *E* anomalies during the drought were mostly positive across the catchments with change (median of 10 %) and negative for catchments with no change (median of -32 %, Fig. 8d). Mean anomalies in annual *P* during the drought were largely negative, and relatively similar for catchments with and without change (Fig. 8d). Mean anomalies in detrended variables ($E_d$ and $P_d$) during the drought were mostly negative, both for catchments with and without change (Fig. S9). The use of $P_{uncorr}$ led to comparable results (Fig. S10).

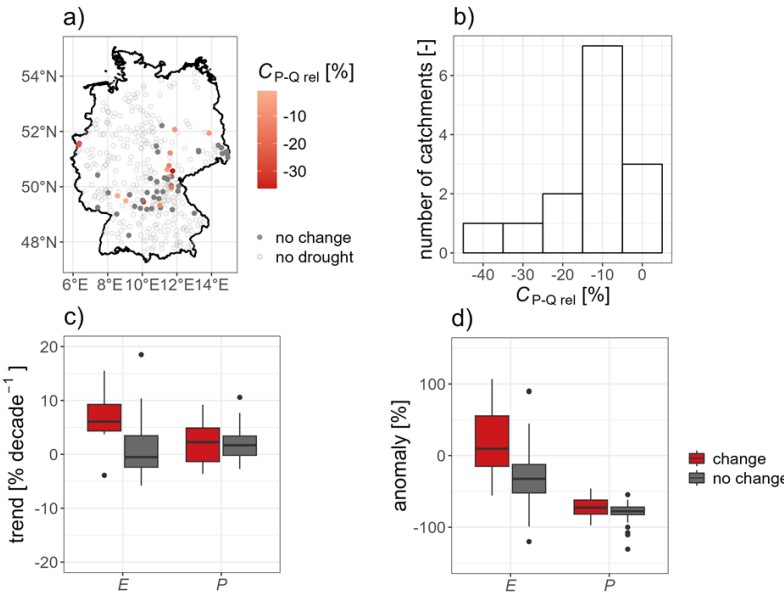

**Fig. 8: Changes in the annual relationship between precipitation (*P*) and streamflow (*Q*, *P-Q* relationship) during the multi-year drought between 1989 and 1993, and their potential predictors. (a) Map of the magnitude of changes in the *P-Q* relationship (*C*<sub>P-Q</sub> rel), across the study catchments. (b) Histogram of *C*<sub>P-Q rel</sub>. (c) Boxplots of trends in annual catchment actual evapotranspiration (*E*) and *P* over 1970–1993, for catchments with change and no change in the *P-Q* relationship. (d) Boxplots of mean anomalies in *E* and *P* over the drought, for catchments with change and no change in the *P-Q* relationship.**

## 4. Discussion

### 4.1 Summer low flows decreased in small catchments in Germany between 1970 and 2019

Trend analysis revealed widespread decreases in summer low flows in all the investigated clusters (Fig. 4, Fig. S3a-d). Summer low flows showed a median trend of -3.7 % decade$^{-1}$ (IQR of -7.5/-0.6 % decade$^{-1}$) across all catchments, with significant negative trends in 31 % of the catchments and significant positive trends in 2 % of them only (Fig. 4b and c). These findings extend for two more decades the decreases in summer low flows which were previously reported for parts of Germany up to the mid-2000s (Stahl et al., 2010; Fangmann et al., 2013). Moreover, our results complement observations of increases in annual low flows by Bormann & Pinter (2017) for catchments with human impacts in Germany between 1950 and 2013. Bormann & Pinter (2017) argued that water management was the main driver of these increases, possibly in combination with climatic changes for catchments with annual *Q* minima in winter. Here we analysed only small catchments without substantial influences of water management, despite human-impacted land cover (Table 1). For these catchments, we observed consistent decreases in summer low flows between 1970 and 2019. Findings of our study and the one by Bormann & Pinter (2017) indicate potential differences in low-flow trends between catchments without and with substantial influences of water management, or between summer and winter low flows, due to different generating processes. The agreement among trends in 7d*Q*<sub>min, JJA</sub> and alternative metrics (i.e., 30d*Q*<sub>min, JJA</sub> and 7d*Q*<sub>min, M-O</sub>, Fig. S4a-f) suggests that our main conclusions remain

valid irrespective of the specific choice of the low-flow metric. However, we found a slightly lower percentage of significant decreases for 30d$Q_{\text{min, JJA}}$ (Fig. S4b) than for 7d$Q_{\text{min, JJA}}$ (Fig. 4b), which may point to stronger potential impacts on river ecosystems than human society. Furthermore, decreases in annual $Q$ and in its timing were more elusive than those in summer low flows (Fig. S4g-l), meaning that summer low flows in particular decreased across the study catchments over 1970–2019, and likely not due to a general decrease in $Q$ or a shift in its timing.

Here we focused on the magnitude of summer low flows. Alternative low-flow metrics also changed in many regions over past decades. For instance, the timing of low flows displayed mostly negative trends (i.e., low flows occurring earlier in the year) in small catchments in Europe up to 2004 (Stahl et al., 2010), and the spatial extent of low flows both increased and decreased across catchments with potentially significant human impacts in Europe between 1969 and 2011 (Brunner and Gilleland, 2021). For a similar set of catchments as the one used here, future work could investigate recent long-term variations in these alternative metrics. Similarly, future studies could focus on streamflow droughts, which are abnormal low flows as compared to the climatology (Van Loon, 2015). Peña-Angulo et al. (2022) found increases in the duration, frequency, and severity of streamflow droughts (i.e., drying conditions over time) in catchments with potentially significant human impacts in Germany between 1962 and 2017. The relevance of water management on these changes might be assessed by selecting catchments with and without substantial influences of water management.

### 4.2 Increases in $E$ were a relevant driver of decreases in summer low flows in small catchments in Germany between 1970 and 2019

The analysis of predictors of decreases in summer low flows revealed concomitant increases in $E$ in all clusters (significant in 27 % of the catchments, Fig. 5b, c, S3e-h), but contrasting changes in seasonal $P$ across the clusters (Fig. 5, S3i-t, Table 2). Specifically, we found that (i) $P_{\text{JJA}}$ decreased in the pre-Alpine and south-central Germany, and increased elsewhere, (ii) $P_{\text{MAM}}$ generally decreased, and (iii) $P_{\text{DJF}}$ generally increased over 1970–2019 (Fig. 5b, e, h), coherently with Duan et al. (2019). These changes indicate that widespread increases in evaporative losses ($E$), local decreases in water input during summer ($P_{\text{JJA}}$), and local decreases in storage recharge during spring (approximated by $P_{\text{MAM}}$) superimposed to generate decreases in summer low flows. These three mechanisms also overcompensated local increases in storage recharge during winter (approximated by $P_{\text{DJF}}$). From a mechanistic point of view, increases in winter storage recharge may also promote increases in vegetation growth and $E$, and thus decreases in soil moisture storage and groundwater recharge during the growing season and ultimately in summer low flows. We speculate here that the negative correlation between trends in $P_{\text{DJF}}$ and in summer low flows in the eastern cluster may suggest this chain of processes (Fig. 7), even though we acknowledge that this relationship may be spurious. Here, we further quantified the relative importance of the considered predictors of changes in summer low flows and showed that the main drivers differed by cluster, i.e., by region with distinct variations in $Q$ over time (Table 2). Increases in $E$ were a significant driver of decreases in 7d$Q_{\text{min, JJA}}$ in the eastern and northern clusters, together with changes in seasonal $P$, whereas decreases in $P_{\text{JJA}}$ dominated the decreases in 7d$Q_{\text{min, JJA}}$ in the pre-Alpine cluster (Fig. 6 and 7). Both multiple linear regressions, focusing on drivers of temporal variations, and the correlation analysis, focusing on drivers of

spatial patterns, agreed on these results. For the south-central cluster, the two analyses led to mixed results which may be due to the generally small trends (Fig. 5). Our results suggest that particular attention should be given to catchments with strong increases in $E$ (e.g., in the eastern and northern Germany, Fig. 5c) to anticipate potential decreases in summer low flows. Moreover, the relevance of increases in $E$ for decreases in summer low flows was strongest for catchments with the highest aridity indices in our sample (i.e., the eastern cluster), despite these catchments are still mostly humid (aridity index < 1, Fig. 3c). Therefore, the aridity index appears as a key attribute to identify catchments where summer low flows are particularly vulnerable to increases in $E$. Our findings regarding the role of increases in $E$ on decreases in summer low flows expand results on the importance of $E$ increases on $Q$ decreases at annual time scale (Fischer et al., 2023; Tran et al., 2023), and findings by Montanari et al. (2023) on concomitant decreases in summer low flows and increases in $E$ in northern Italy over recent decades. Furthermore, our results align with those of Thomas et al. (2015), Fangmann & Haberlandt (2019), and Floriancic et al. (2021) on the importance of potential evapotranspiration for the interannual variability of low flows in German catchments, and those of Hammond et al. (2022) on decreases in low flows in catchments in the USA where aridity increased. Increases in $E$ showed a halt in Germany after the 2000s (Fig. 4a and Bruno & Duethmann, 2024), leading to milder trends in $E$ over 1970–2019 than over 1970–1999 (Fig. S5). Increases in $E$ over 1970–1999 correlated with decreases in summer low flows in the pre-Alpine cluster as well (Fig. S8), suggesting that the contribution of increases in $E$ to decreases in summer low flows for regions and periods with monotonic $E$ increases may be even larger than the one that we illustrated here.

**Table 2: Summary of long-term variations in summer low flows and in their potential predictors (annual evapotranspiration, $E$, and precipitation over summer, $P_{JJA}$, spring, $P_{MAM}$, and winter, $P_{DJF}$) over 1970–2019, and drivers of variations in summer low flows (only significant ones according to both attribution analyses, Sect. 2.5) for each cluster. Red arrows refer to strong decreases, light red arrows to mild decreases, light blue arrows to mild increases, and blue arrows to strong increases as median across the catchments, with ± 2 % decade$^{-1}$ as thresholds for strong increases/decreases.**

| | Cluster | | | |
|---|---|---|---|---|
| | Pre-Alpine | South-Central | Eastern | Northern |
| Variations in summer low flows | ↓ (light red) | ↓ (light red) | ↓ (red) | ↓ (light red) |
| Variations in $E$ | ↑ (light blue) | ↑ (light blue) | ↑ (blue) | ↑ (blue) |
| Variations in $P_{JJA}$ | ↓ (light red) | ↓ (light red) | ↑ (blue) | ↑ (light blue) |
| Variations in $P_{MAM}$ | ↑ (light blue) | ↓ (light red) | ↓ (light red) | ↓ (red) |
| Variations in $P_{DJF}$ | ↑ (blue) | ↑ (blue) | ↑ (blue) | ↑ (blue) |
| Drivers of variations in summer low flows | $P_{JJA}$ | - | $E$ | $E$, $P_{JJA}$ |

High $E$ contributed to streamflow droughts in the Alps during the warm and dry summer 2003 (Mastrotheodoros et al., 2020), and delayed the recovery of $S$ deficits during the 2018–2021 drought in Germany (Boeing et al., 2024). However, the effect of increases in $E$ on potential increases in streamflow drought conditions remains unclear. Future research could quantify it, similarly to the analyses by Lan et al. (2024) to disentangle the role of vegetation greening on increases in streamflow drought conditions in China between 1982 and 2015.

## 4.3 During the multi-year drought in Germany in the early 1990s, the occurrence of changes in the annual *P-Q* relationship was related with increases in *E* over past decades

Changes in the annual *P-Q* relationship during the multi-year drought in Germany in the early 1990s generally occurred in catchments with increases in *E* (Fig. 8). Following Saft et al. (2015) and Massari et al. (2022), we found that 15 % of the study catchments underwent a multi-year drought between 1989 and 1993, and 26 % of these catchments showed a change in the *P-Q* relationship, with respect to the one before the drought (Fig. 8a). For catchments with such a change, we found a median decrease in *Q* of -10 % compared to the *P-Q* relationship before the drought. Thus, these catchments generated less *Q* during the multi-year drought than was predicted by their *P-Q* relationship before the drought and the observed *P*. These changes should not be seen as long-term shifts in the hydrological regime of the catchments, since we did not include years after the drought in the analysis in order to exclude potential non-recovery from the drought conditions for possibly different processes (Peterson et al., 2021). According to this analysis, the multi-year drought in Germany in the early 1990s had less severe impact on *Q* generation than the Millennium drought in Australia (changes in 56 % of the catchments, with median decrease of approximately -50 %, Saft et al. 2015), the 2012–2016 event in California (mean decreases of -28 % across three catchments, Avanzi et al. 2020), and the 2010–2020 drought in Chile (changes in 61 % of the catchments and mean decrease of -19 %, Alvarez-Garreton et al., 2021). These differences may be related to the characteristics of the *P* anomalies (severity and duration), potential data uncertainties (Sect. 4.4), and the hydro-climatic characteristics of the catchments. For catchments in Europe and droughts of similar duration to the one that we analysed here, Massari et al. (2022) reported changes in 33 % of the catchments, and a median decrease of -28 % across catchments and droughts. Our findings, therefore, confirm less widespread and severe changes in the *P-Q* relationship for specific droughts in Europe than for other events in different hydro-climatic settings (e.g., the Millennium drought in Australia). Massari et al. (2022) suggested positive *E* anomalies during the multi-year droughts as causes of these changes in Europe. We revealed that catchments with changes during the drought predominantly had strong increases in *E* before and during the drought, while catchments with no change mostly had negligible changes in *E* over the same period (Fig. 8c). Furthermore, catchments with change in the *P-Q* relationship had mostly positive mean anomalies in *E* (Fig. 8d), but not in detrended *E* (Fig. S9). Our results point out that long-term increases in *E* between the 1970s and 1990s were related with positive *E* anomalies during the multi-year drought in Germany in the early 1990s and thus, led to decreases in *Q* compared to the *P-Q* relationship before the drought. Monitoring long-term variations in *E*, therefore, can help identifying catchments potentially prone to alterations in *Q* generation during extended dry periods. We also found that catchments with and without change in the *P-Q* relationship had comparable *P* trends (mild long-term increases in *P* before and during the drought, Fig. 8c) and comparable *P* anomalies (negative *P* anomalies of similar magnitude, Fig. 8d) during the analysed multi-year drought. In other words, the catchments with changes in the *P-Q* relationship were not characterized by more severe *P* deficits or drying trends than those with no change. This reinforces the importance of increases in *E* as a predictor of changes in the *P-Q* relationship during the analysed multi-year drought.

Additional processes, such as decreases in the contribution of *S* to *Q* under prolonged dryness, may have further contributed to changes in the *P-Q* relationship during this event, as reported for the Millennium drought in south-eastern Australia (Trotter

et al., 2024; Gardiya Weligamage et al., 2023). Future research might investigate the extent of changes in the *P-Q* relationship during other multi-year droughts in Germany over recent decades, such as the 2018–2022 event (Boeing et al., 2024; Sodoge et al., 2024) only partly covered by our dataset, and their causes.

## 4.4 Sources of uncertainty

We opted for a data-based analysis, given the current issues of hydrological models in representing low flows (Staudinger et
al., 2011), long-term variations (Duethmann et al., 2020), and hydrological changes during prolonged dry periods (Fowler et al., 2020). Measuring $Q$ during dry periods is unavoidably challenging (Coxon et al., 2015), but long-term errors in the same direction can be assumed unlikely across a large number of catchments (Duethmann et al., 2020). Uncertainties in $P$ can arise from potential inhomogeneities in gauge data over time and gauge undercatch. Here, we used a gridded $P$ dataset from the interpolation of a fixed number of gauges over time to minimize inhomogeneities and we corrected it for gauge undercatch
(Sect. 2.2). This correction procedure may lead to further uncertainties, but we achieved comparable results for all analyses for both $P$ and $P_{uncorr}$ (Sect. 3.2 and 3.3), meaning that our conclusions do not depend on the use of the correction procedure. Given the generally coarse resolution of satellite-derived $E$ estimates, we computed $E$ from observed $P$ and $Q$, and estimates of $S_{dyn}$ of the catchments from $Q$ data, as a first order approximation of $S$. This approach may be problematic for specific catchments, such as those with relevant intercatchment groundwater flows (Fan, 2019; Kampf et al., 2020; Safeeq et al., 2021).
Yet, previous works often used a water balance-approach to study long-term variations in $E$ (Teuling et al., 2009; Ukkola and Prentice, 2013; Duethmann and Blöschl, 2018) and Bruno and Duethmann (2024) showed that long-term variations of water balance-derived $E$ were generally robust to uncertainties in $P$ and $S$ for small catchments in Germany over the last five decades, and coherent with those from point-scale $E$ data. Here, we further aimed at minimizing uncertainties in $E$ for specific catchments and years by using cluster-, 5-year averages in the multiple linear regressions for trend attribution (Sect. 2.5).
Finally, as potential predictors of changes in summer low flows we approximated storage processes with $P$ in the season preceding summer, due to unavailability of long-term $S$ data for the study catchments. We chose this approach instead of using alternative proxies for $S$ (e.g., estimates of $S_{dyn}$ or baseflow from $Q$ data) to avoid dependences between predictors and target variable (summer low flows). The satisfactory performances of the multiple linear regressions and the plausible signs of their coefficients suggest the suitability of the selected predictors to represent the long-term dynamic of summer low flows (Table
S2 and Fig. S7a and b).

## 4.5 Implications

Summer is the season when $Q$ is most critical for river ecosystems and specific human uses, such as irrigation. Multi-year droughts can have different impacts on ecosystems and societies than single dry years, as shown for the 2018–2022 drought in Germany which had stronger effects on forestry, recreational activities, aquaculture, and waterborne transportation than the
2003 and 2015 events, with impacts mostly on agriculture (Sodoge et al., 2024). Thus, long-term decreases in summer low

flows and stronger reductions in $Q$ than expected during multi-year droughts can have direct ecological and socio-economic implications, and a proper understanding of the causes of these phenomena is required to better predict them.

Here, we revealed that increases in $E$ contributed (i) to decreases in summer low flows, in particular in catchments with relatively higher aridity indices, and (ii) to changes in the $P$-$Q$ relationship during the multi-year drought that occurred in Germany between 1989 and 1993. With strong increases in $E$ reported for many regions of the world over recent decades (Duethmann and Blöschl, 2018; Teuling et al., 2019; Yang et al., 2023), these findings are highly relevant beyond our study region.

Under further projected future increases in $E$ (Yang et al., 2023) and multi-year droughts in many regions, as shown by Van Der Wiel et al. (2023) for the Rhine basin for instance, we underline the importance of monitoring changes in $E$ for the prediction of potential decreases in $Q$ during dry periods, particularly in arid regions. Furthermore, given the frequent challenges of hydrological models under changing conditions (Saft et al., 2016) and in representing long-term variations in $E$ (Duethmann et al., 2020), we recommend evaluating models with particular attention to long-term variations in $E$, and possibly improving them, when used for climate impact assessments on river ecosystems (Barbarossa et al., 2021) and human societies (Naumann et al., 2021).

## 5. Conclusions

The magnitude of summer low flows widely decreased across 363 small catchments without substantial influences of water management in Germany over 1970–2019, with significant negative trends in 31 % of the catchments (Fig. 4). Long-term increases in $E$ were a relevant driver of decreases in summer low flows, particularly for humid-to-arid eastern catchments, both in terms of contribution to temporal dynamics (35 %, Fig. 6) and coherence of spatial patterns ($r$ equal to -0.74, Fig. 7). A change in the $P$-$Q$ relationship occurred in 26 % of the catchments with a multi-year drought in the early 1990s, with lower $Q$ than expected from the relationship before the drought. Catchments with a change were characterized by strong underlying increases in $E$ (median trend of 6.1 % decade$^{-1}$), while catchments without change had on average negligible changes in $E$ (median trend -0.5 % decade$^{-1}$; Fig. 8). We illustrated the imprint of long-term increases in $E$ on decreases in $Q$ during dry conditions, which can be especially relevant for arid regions and prolonged droughts. Therefore, considering long-term variations in $E$ is essential to understand and model changes in $Q$ during dry periods, for the adaptation of water management to global changes.

**Code Availability**

Codes will be made available upon request.

## Data Availability

Streamflow data and catchment boundaries are available through the Environment Agencies of German Federal States (Landesamt für Umwelt Brandenburg; Landesanstalt für Umwelt Baden-Württemberg; Bayerisches Landesamt für Umwelt; Landesamt für Umwelt, Naturschutz und Geologie Mecklenburg-Vorpommern; Niedersächsischer Landesbetrieb für Wasserwirtschaft, Küsten- und Naturschutz; Landesamt für Natur, Umwelt und Verbraucherschutz Nordrhein-Westfalen; Landesamt für Umwelt Rheinland-Pfalz; Landesamt für Landwirtschaft, Umwelt und ländliche Räume Schleswig-Holstein;

Landesamt für Umwelt- und Arbeitsschutz Saarland; Sächsisches Landesamt für Umwelt, Landwirtschaft und Geologie; Landesbetrieb für Hochwasserschutz und Wasserwirtschaft Sachsen-Anhalt; und Thüringer Landesamt für Umwelt, Bergbau und Naturschutz).

## Author contribution

GB: conceptualization; methodology; software; validation; formal analysis; investigation; data curation; writing – original
draft; writing – review and editing; visualization. LS: software; writing – review and editing. DD: methodology; software; formal analysis; investigation; writing – review and editing; supervision; project administration; funding acquisition.

## Competing interests

The authors declare that they have no conflict of interest.

## Acknowledgments

We thank Peter Hoffmann for providing the precipitation dataset, the Environment Agencies of German Federal States for streamflow data and catchment boundaries, and the Editor and Reviewers for their constructive comments to improve the manuscript.

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
