# Peer review of "Imprints of Increases in Evapotranspiration on Decreases in Streamflow during Dry Periods, a Large-sample Analysis in Germany"

_EGUsphere, 2024_

## Author Comment (AC1)

Dear Referee,

we thank you for your time in reviewing our manuscript and your constructive feedback. We see feasible implementing all your suggestions in a revised version of the manuscript. In particular, we have now performed the multiple linear regression analysis also at the catchment scale, which we plan to introduce in a revised version of the manuscript. We also intend discussing more the reasons for our methodological choices and potential uncertainties in the study.

Please find below our point-by-point reply to your comments (italic, numbered) and the changes we propose to do in the manuscript to address them (underlined).

Best regards,

Giulia Bruno and co-authors

*This paper analysed the summer low flows by using the datasets from 363 small catchments. The relationships among evapotranspiration (E), precipitation (P) and streamflow (Q), as well as the storage (S) were quantified. Results showed that summer low flows decreased significantly, and increased E played the main driver in the eastern catchments. In addition, the P-Q relationship changed in 26% of catchments between 1989 and 1993. Generally, the structure of the paper is clear and well-organized, however, there are some concerns.*

We thank you for appreciating the clarity of our manuscript and for raising interesting points to improve our work.

*Specific comments:*

1. *The coefficients of a and b in equation (2) should be vary with wind direction and elevation of gauges. Does authors calculate them for each gauges? If so, please provide the analysis which are the main uncertainty for precipitation datasets.*

The method we used for the correction of precipitation (*P*) for gauge undercatch was developed for point data (Richter, 1995), with coefficients depending on gauge characteristics (wind exposure) and meteorological conditions (*P* type, Eq. 2 in the manuscript). Here, we used a gridded *P* dataset (Sect. 2.2) and we applied the correction procedure at pixel-scale, by assuming all pixels as moderately sheltered with respect to wind exposure. This approach was used also in previous works (Duethmann & Blöschl, 2018; Bruno & Duethmann, 2024). Duethmann & Blöschl (2018) showed that alternative assumptions regarding the corrective coefficients had small influences on long-term variations in *P* and water-balance derived catchment evapotranspiration (*E*), we are mostly interested in for our analyses. Thus, we argue that this assumption unlikely affected our main conclusions. We agree, however, that *P* data are unavoidably affected by some degree of uncertainty, either per se and following this correction, which we do not discuss in the manuscript at the moment. We propose to modify L134–137 to make clearer our assumption regarding the correction procedure and to add some discussion concerning the uncertainty in *P* data in a new section (4.4 Sources of uncertainty).

L134–137: We corrected the dataset for gauge undercatch following the method proposed for Germany by Richter (1995):

$$P_{corr} = P_{uncorr} + aP_{uncorr}{}^{b}$$

(2)

with $P_{corr}$ as corrected $P$, $P_{uncorr}$ uncorrected $P$, $a$ and $b$ coefficients which vary with wind exposure of the gauges, precipitation type (rain or snow), and season. Here, we assumed for all grid cells the coefficients for moderately sheltered locations in Richter (1995), given the low sensitivity of long-term variations in $P$ to the selected coefficients (Duethmann & Blöschl, 2018).

4.4     Sources of uncertainty: Uncertainties in $P$ can arise from potential inhomogeneities in gauge data over time and gauge undercatch. Here, we used a gridded $P$ dataset from the interpolation of a fixed number of gauges over time to minimize inhomogeneities and we corrected it for gauge undercatch (Sect. 2.2). This correction procedure may lead to further uncertainties, but Duethmann and Blöschl (2018) demonstrated that assumptions regarding corrective coefficients have little influence on trends in $P$ and $E$, we were mostly interested in here.

> 2.  *It is quite difficult to understand the equation (4), where the dynamics of storage was approximated by P_mam and P_djf. Please provide more explanations. In addition, since baseflow is 0.66 in this area, which means the soil moisture and groundwater both plays important roles in runoff variation. But it seems they are not taken into account in the analysis.*

We agree that soil and groundwater storage are relevant for $Q$ generation in the study catchments. As predictors of trends in summer low flows, we indeed considered variations in $E$, summer $P$, and storage ($S$). We accounted for the influence of $S$ variations using winter precipitation ($P_{DIF}$) and spring precipitation ($P_{MAM}$) as wetness conditions in the seasons preceding the dry period in the study region and therefore, proxies for $S$ recharge. We used these proxies since long-term data on soil moisture and groundwater are unavailable for the large sample of small catchments that we analyzed, similarly to what done by previous works (Duethmann et al., 2015; Saft et al., 2016; Laaha et al., 2017). We propose to rephrase L161–162 as follows and to discuss this point in the new section 4.4.

L161–162: Since long-term data on soil moisture and groundwater storage are not available for the study catchments, we used $P_{MAM}$ and $P_{DJF}$ as proxies of storage recharge in the seasons preceding the dry one (Duethmann et al., 2015; Saft et al., 2016; Laaha et al., 2017).

4.4 Sources of uncertainty: Finally, as potential predictors of changes in summer low flows we approximated storage processes with $P$ in the season preceding the dry one, due to unavailability of long-term $S$ data for the study catchments. We chose this approach instead of using alternative proxies for $S$ (e.g., estimates of $S_{dyn}$ or baseflow from $Q$ data) to avoid dependences between predictors and target variable (summer low flows). The satisfactory performances of the multiple linear regressions and the plausible signs of their coefficients suggest the suitability of the selected predictors to represent the long-term dynamic of summer low flows (Table S1).

> 3.  *For multiple linear regression in predicting summer low flows, the authors showed the R2 in four clusters which exhibited good performance in Table S1. However, there is spatial variation among different gauges so the coefficients vary at each gauge, does the regression in the gauge scale follow the same trend with cluster?*

In a first step, we performed the multiple linear regression at a cluster-scale to minimize uncertainties in $E$ for specific catchments (see also reply to comment #4 by Referee #2). However, we see that such analysis may not fully reveal potential spatial differences in the predictors of the temporal dynamics of summer low flows. Thus, we have now repeated the analysis at the catchment-scale. We achieved overall satisfactory results in terms of model performances (median coefficient of determination across the catchments equal to 0.78, Fig. 1a and b here) and in line with those at a cluster-scale. By looking at the predictor with highest contribution to the simulation (primary predictor) for each catchment, we found that summer precipitation ($P_{JJA}$) was the most recurrent one across all clusters (Fig. 1c). $P_{JJA}$ was frequently non-significant though, especially where model performances were

relatively low (Fig. 1a) and predictors had a similar relative contribution (not shown). By focusing on significant primary predictors only, the most recurrent ones were $P_{JJA}$ in the Pre-Alpine cluster, $P_{MAM}$ in the South-Central one, and $E$ in the Eastern and Northern clusters (Fig. 1d). These are coherent with the conclusions we draw at the cluster-scale (i.e., $P_{JJA}$ dominant predictor in the Pre-Alpine cluster, $P_{MAM}$ significant predictor in the South-Central one, and $E$ in the Eastern and Northern clusters, Fig. 6 in the manuscript). To reinforce our trend attribution, we intend to add this analysis in a revised version of the manuscript, by introducing Fig. 1 as a new Fig. S6 in the Supplement, and adapting the description of methods and results as follows.

[Figure]

Fig. 1: Model results from the multiple linear regressions on catchment-scale data. (a) Map of coefficients of determination of the models ($R^2$). (b) Histogram of $R^2$. (c) Map of primary predictors (black edges if significant). (d) Relative frequency of primary predictors by cluster. Light grey in (a, c, and d) refers to catchments with high multicollinearity of the predictors, and thus excluded from the analysis (Sect. 2.5). In (d), Pre-al. refers to Pre-Alpine, South. to South-Central, East. to Eastern, and North. to Northern cluster.

Methods: Firstly, we modelled the temporal dynamics of summer low flows ($7dQ_{min, JJA}$ in Eq.(4)) both at a catchment- and cluster-scale from the dynamics of the predictors through multiple linear regression:

$$7dQ_{min, JJA} = \alpha_1 E + \alpha_2 P_{JJA} + \alpha_3 P_{MAM} + \alpha_4 P_{DJF} + \varepsilon \qquad (4)$$

with $\alpha_i$ ($i$ = 1…4) the regression coefficient for each predictor and $\varepsilon$ the model residuals. We adopted 5-year averages to focus on long-term dynamics and reduce uncertainties in water balance-derived $E$. Moreover, for the cluster-scale analysis we used average time series across the catchments in each cluster to minimize uncertainties in $E$ for specific catchments, while analysing the main signal at a regional scale.

Results: Multiple linear regression for predicting long-term dynamics in summer low flows achieved satisfactory performances both at cluster- and catchment-scale (for $7dQ_{\text{min, JJA}}$, $R^2 > 0.7$ for each cluster and median $R^2$ of 0.78 across all catchments, Table S1, and Fig. S6a and b).

Catchment-scale results showed similar patterns, despite being unavoidably more affected by noise than those at the cluster-scale (Fig. S6c and d). By focusing on significant primary predictors only (i.e., predictors with highest contribution to the simulations), $P_{\text{JJA}}$ was the most recurrent one in the Pre-Alpine cluster, $P_{\text{MAM}}$ in the South-Central cluster, and $E$ in the Eastern and Northern clusters (Fig. S6d).

4. *Could authors discuss why the r= -0.49 in P_djf for the easter cluster in lines 258?*

In the Eastern cluster, we indeed detected a negative correlation between trends in summer low flows and trends in $P_{\text{DJF}}$, which may sound counterintuitive under our assumption of $P_{\text{DJF}}$ as a proxy for storage recharge during the wet season. While we acknowledge that spurious effects in the correlation analysis may play a role here, we see a mechanistic explanation, related to $E$-storage feedbacks (Boeing et al., 2024) and the observed changes in this cluster. In particular, catchments in the Eastern cluster showed both positive and negative trends in $P_{\text{DJF}}$, but generally negative trends in $P_{\text{MAM}}$ and positive trends in $E$. This means that increases in $P_{\text{DJF}}$, and thus in storage conditions at the beginning of the growing season, might have buffered the decreases in $P_{\text{MAM}}$ in sustaining the increases in $E$ in some catchments. Increases in $E$, in turn, contributed to decreases in summer low flows in this cluster (Fig. 6 and 7 in the manuscript). Thus, increases in $P_{\text{DJF}}$ may have indirectly contributed to decreases in summer low flows in the Eastern cluster. We propose to expand the mechanistic explanation on this point as follows.

These three mechanisms (i.e., widespread increases in $E$, local decreases in $P_{\text{JJA}}$, and local decreases in $P_{\text{MAM}}$, in the previous sentence not reported here) also overcompensated local increases in storage recharge during winter (approximated by $P_{\text{DJF}}$), possibly through $E$-storage feedbacks (Boeing et al., 2024) for instance in the Eastern cluster (Fig. 7).

5. *For 363 catchments, there is only 15 catchment with negative Cp-q rel values which distributed sparsely in Fig.8(a). I am curious about the possibility caused by data process uncertainty.*

Having no changes in the *P-Q* relationships even during prolonged dry periods was common expectation for humid catchments until recently (Massari et al., 2022) and a sparse occurrence of these changes is positive for water management (see e.g. Fowler et al., 2022 for practical implications of these changes). We see that our study revealed less widespread changes in the *P-Q* relationship during the multi-year drought in Germany in the early 1990s than previous works for other case studies. We discuss that these differences may relate to the characteristics of the multi-year droughts (i.e., severity and duration of the *P* deficits) and to the hydro-climatic properties of the catchments. We agree that the unavoidable uncertainty in *P* and *Q* data (see reply to comment #1) may also play a role, despite we here aimed at minimizing uncertainty in *P* data by using a dataset specifically tailored to long-term consistency (Sect. 2.2). To add this point to the Discussion, we propose to rephrase L360–365 as follows.

According to this analysis, the multi-year drought in Germany in the early 1990s had less severe impact on *Q* generation than the Millennium drought in Australia (changes in 56 % of the catchments, with median decrease of approximately -50 %, Saft et al. 2015), the 2012–2016 event in California (mean decreases of -28 % across three catchments, Avanzi et al. 2020), and the 2010–2020 drought in Chile (changes in 61 % of the catchments and mean decrease of -19 %, Alvarez-Garreton et al., 2021). These differences may be related to the characteristics of the *P* anomalies (severity and duration), potential

uncertainties in the underlying data (Sect. 4.4), and the hydro-climatic characteristics of the catchments.

6. *For Fig.5 and Table S1, it showed that P_jja played a major contribution to Q changes in Pre-Alpine and South-Central cluster, while E played much more contribution in Eastern and Northern area. Does it relate to elevation changes?*

Catchments in the Pre-Alpine and South-Central clusters have indeed relatively higher mean elevations than others (Fig. 3 in the manuscript). Furthermore, catchments in the Pre-Alpine cluster generally experienced decreases in $P_{JJA}$ and mild increases in $E$. Catchments in the South-Central cluster showed similar behaviors, but overall small trends. On the contrary, catchments in the Eastern and Northern clusters largely had increases in $P_{JJA}$ and $E$. Therefore, we argue that differences in the main drivers of decreases in summer low flows between the Pre-Alpine and North-Eastern areas can be ascribed to differences in the variations in the drivers themselves, rather than to differences in catchment characteristics like elevation. To make the differences in hydro-climatic changes among the clusters easier to grasp, we propose to add the following summary table in Section 4.2.

Table 1: Summary of long-term variations in summer low flows and in their potential predictors (annual evapotranspiration, $E$, and precipitation over summer, $P_{JJA}$, spring, $P_{MAM}$, and winter, $P_{DJF}$) over 1970–2019, and drivers of variations in summer low flows (only significant ones according to both attribution analyses, Sect. 2.5) for each cluster. Red arrows refer to strong decreases, light red arrows to mild decreases, light blue arrows to mild increases, and blue arrows to strong increases at the median level, with ± 2 % decade$^{-1}$ as thresholds for strong increases/decreases.

| | Cluster | | | |
|---|---|---|---|---|
| | Pre-Alpine | South-Central | Eastern | Northern |
| Variations in summer low flows | ↓ | ↓ | ↓ | ↓ |
| Variations in $E$ | ↑ | ↑ | ↑ | ↑ |
| Variations in $P_{JJA}$ | ↓ | ↓ | ↑ | ↑ |
| Variations in $P_{MAM}$ | ↑ | ↓ | ↓ | ↓ |
| Variations in $P_{DJF}$ | ↑ | ↑ | ↑ | ↑ |
| Drivers of variations in summer low flows | $P_{JJA}$ | - | $E$ | $E$, $P_{JJA}$ |

References:

Alvarez-Garreton, C., Boisier, J. P., Garreaud, R., Seibert, J., & Vis, M. (2021). Progressive water deficits during multiyear droughts in basins with long hydrological memory in Chile. *Hydrology and Earth System Sciences*, *25*(1), 429–446. https://doi.org/10.5194/hess-25-429-2021

Avanzi, F., Rungee, J., Maurer, T., Bales, R., Ma, Q., Glaser, S., & Conklin, M. (2020). Climate elasticity of evapotranspiration shifts the water balance of Mediterranean climates during multi-year droughts. *Hydrology and Earth System Sciences*, *24*(9), 4317–4337. https://doi.org/10.5194/hess-24-4317-2020

Boeing, F., Wagener, T., Marx, A., Rakovec, O., Kumar, R., Samaniego, L., & Attinger, S. (2024). Increasing influence of evapotranspiration on prolonged water storage recovery in Germany. *Environmental Research Letters*, *19*(2), 024047. https://doi.org/10.1088/1748-9326/ad24ce

Bruno, G., & Duethmann, D. (2024). Increases in Water Balance-Derived Catchment Evapotranspiration in Germany During 1970s–2000s Turning Into Decreases Over the Last Two Decades, Despite Uncertainties. *Geophysical Research Letters*, *51*(6), e2023GL107753. https://doi.org/10.1029/2023GL107753

Duethmann, D., & Blöschl, G. (2018). Why has catchment evaporation increased in the past 40 years? A data-based study in Austria. *Hydrology and Earth System Sciences*, *22*(10), 5143–5158. https://doi.org/10.5194/hess-22-5143-2018

Duethmann, D., Bolch, T., Farinotti, D., Kriegel, D., Vorogushyn, S., Merz, B., Pieczonka, T., Jiang, T., Su, B., & Güntner, A. (2015). Attribution of streamflow trends in snow and glacier meltdominated catchments of the Tarim River, Central Asia. *Water Resources Research*, *51*(6), 4727–4750. https://doi.org/10.1002/2014WR016716

Fowler, K., Peel, M., Saft, M., Nathan, R., Horne, A., Wilby, R., McCutcheon, C., & Peterson, T. (2022). Hydrological Shifts Threaten Water Resources. *Water Resources Research*, *58*(8), Article 8. https://doi.org/10.1029/2021WR031210

Laaha, G., Gauster, T., Tallaksen, L. M., Vidal, J.-P., Stahl, K., Prudhomme, C., Heudorfer, B., Vlnas, R., Ionita, M., Van Lanen, H. A. J., Adler, M.-J., Caillouet, L., Delus, C., Fendekova, M., Gailliez, S., Hannaford, J., Kingston, D., Van Loon, A. F., Mediero, L., … Wong, W. K. (2017). The European 2015 drought from a hydrological perspective. *Hydrology and Earth System Sciences*, *21*(6), 3001–3024. https://doi.org/10.5194/hess-21-3001-2017

Massari, C., Avanzi, F., Bruno, G., Gabellani, S., Penna, D., & Camici, S. (2022). Evaporation enhancement drives the European water-budget deficit during multi-year droughts. *Hydrology and Earth System Sciences*, *26*(6), 1527–1543. https://doi.org/10.5194/hess-26-1527-2022

Richter, D. (1995). *Ergebnisse methodischer Untersuchungen zur Korrektur des systematischen Messfehlers des Hellmann- Niederschlagsmessers*. *Offenbach*.

Saft, M., Peel, M. C., Western, A. W., & Zhang, L. (2016). Predicting shifts in rainfall-runoff partitioning during multiyear drought: Roles of dry period and catchment characteristics. *Water Resources Research*, *52*(12), 9290–9305. https://doi.org/10.1002/2016WR019525

Saft, M., Western, A. W., Zhang, L., Peel, M. C., & Potter, N. J. (2015). The influence of multiyear drought on the annual rainfall-runoff relationship: An Australian perspective. *Water Resources Research*, *51*(4), 2444–2463. https://doi.org/10.1002/2014WR015348

---

## Author Comment (AC2)

Dear Referee,

we thank you for your time in reviewing our manuscript and your constructive feedback. We see feasible implementing all your suggestions in a revised version of the manuscript. In particular, we have now assessed long-term variations in additional streamflow metrics and carefully evaluated our approach for the estimation of the dynamic storage of the catchments. We plan to introduce these additional analyses in a revised version of the manuscript. We also intend discussing more the reasons for our methodological choices and potential uncertainties in the study.

Please find below our point-by-point reply to your comments (italic) and the changes we propose to do in the manuscript to address them (underlined).

Best regards,

Giulia Bruno and co-authors

*Bruno et al. analyzed the decreasing trend of low flow in 363 small catchments in Germany. They also attributed the decrease to the increase of ET. They further unraveled that the change of P-Q relationship during drought produces lower flow which is generally due to the increased ET. They conducted this work based on observations of P and Q, empirical expression of subsurface storage, and ET derived based on water balance and statistical analysis. I think studying the decrease of low flow and trying to find the major drivers is very important in the climate change background. The data and analysis are generally reliable, the structure and the writing are good. However, there the following concerns which need clarification from authors for further review.*

We thank you for your overall positive evaluation of our manuscript and for raising interesting points to improve our work.

1. *I was more or less confused by the overall idea of the authors. If you wanted to check if the decrease of the flow is caused by increased ET, and you also have calculated the water balance, why not do a straightforward analysis of the overall change of P, ET, Q, and S. Then it is easy to get if the decrease of flow is mainly driven by ET. Then you can do the analysis in your manuscript as a follow-up. Otherwise, I feel the conclusions are even not that convincing as, for example, the decrease of low flow might be just because of the shift of the timing of streamflow.*

We understand that you suggest evaluating here the impact of changes in catchment evapotranspiration ($E$) on streamflow ($Q$) at an annual time scale. This has already been well documented for several regions and periods, partly overlapping with our case study as well (Teuling et al., 2009; Fischer et al., 2023; Renner & Hauffe, 2024). Here, we rather aimed at assessing the impact of increases in $E$ on $Q$ during dry periods specifically, to complement these previous findings. We see, however, that assessing changes in additional $Q$ metrics, such as annual $Q$ and its timing, may provide useful context regarding hydrological changes in the region and the decreases in the magnitude of summer low flows that we observed. Therefore, we have now quantified changes in annual $Q$ and its timing, the latter through the center of mass date of $Q$ ($CMD_Q$, Court, 1962; Han et al., 2024). Annual $Q$ and $CMD_Q$ generally showed decreases across the catchments between 1970 and 2019 (Fig. 1 and 2 here). However, these decreases were less significant than those in the magnitude of summer low flows, with a median trend in annual $Q$ ($CMD_Q$) of -1.3 % decade$^{-1}$ (-0.006 month year$^{-1}$) across all catchments and significant negative trends in 10 (3) % of them, as compared to a median trend in

7d$Q_{min, JJA}$ of -3.7 % decade$^{-1}$ across all catchments and significant negative trends in 31 % (see Fig. 1 and 2 here for annual $Q$ and CMD$_Q$, and Fig. 4 in the manuscript for 7d$Q_{min, JJA}$).

[Figure]

Fig. 1: Long-term variations in annual streamflow ($Q$) over 1970–2019. (a) Average anomalies across the study catchments. (b) Map of catchment-scale trends (black edges if significant). (c) Boxplots of trends for all catchments and by cluster.

[Figure]

Fig. 2: Long-term variations in the centre of mass date of streamflow (CMD$_Q$) over 1970–2019. (a) Average anomalies across the study catchments. (b) Map of catchment-scale trends (black edges if significant). (c) Boxplots of trends for all catchments and by cluster.

Therefore, summer low flows in particular decreased across the study catchments between 1970 and 2019, as compared to other $Q$ metrics, and these decreases did not occur simply with an overall decrease in $Q$ or a shift in its timing. We plan to introduce this analysis in a revised version of the manuscript, by adding Fig. 1 and 2 in the Supplement, and methodological aspects, results, and discussion of this new analysis in the corresponding sections.

Methods: To characterize general long-term variations in $Q$, we also considered annual $Q$ and the center of mass date of $Q$ (CMD$_Q$). CMD$_Q$ is the time of the year in which half of the annual $Q$ occurs and as such, a metric for $Q$ timing (Court, 1962; Han et al., 2024).

Results: Additional $Q$ metrics similarly decreased over 1970–2019, even though less significantly than summer low flows (median trend in annual $Q$ of -1.3 % decade$^{-1}$ across all the catchments and significant negative trends in 10 % of them, and median trend in CMD$_Q$ of -0.006 month year$^{-1}$ across all the catchments and significant negative trends in 3 % of them).

Discussion: Furthermore, decreases in annual $Q$ and in its timing (CMD$_Q$) were more elusive than those in summer low flows, meaning that summer low flows in particular decreased across the study catchments over 1970–2019, and likely not simply due to a general decrease in $Q$ or a shift in its timing.

Furthermore, we intend adding more context around current knowledge on the effect of increases in $E$ on decreases in $Q$ at an annual time scale in the Introduction and Discussion as follows, also to highlight better the novelty of our study with respect to previous literature.

Introduction: Furthermore, increases in *E* contributed to decreases in annual *Q* on the long-term (Teuling et al., 2009; Fischer et al., 2023; Renner & Hauffe, 2024) and during dry years (Tran et al., 2023), especially in mountain catchments.

Discussion: Our findings on the role of increases in *E* on decreases in summer low flows expand results at annual time scale (Fischer et al., 2023; Tran et al., 2023), and those by Montanari et al. (2023) on concomitant decreases in summer low flows and increases in *E* in northern Italy over recent decades.

> 2. *You mentioned you used Kirchner's approach to calculate S$_{dyn}$ which needs that S is the main control of Q generation. I am wondering if the 363 catchments you used meet this requirement and where is your analysis for this?*

We agree that the assumption of *Q* mainly controlled by the dynamic storage ($S_{dyn}$) may not always hold true across large samples of catchments, despite previous studies adopted it (Staudinger et al., 2017; Trotter et al., 2024) also in our study region (Stoelzle et al., 2013; Berghuijs et al., 2016). We have now evaluated this assumption similarly to what done by Kirchner (2009). Specifically, we computed the percentage of winter days (November to April, included) with (i) rising *Q* (d*Q*/dt > 0) when precipitation (*P*) exceeds *Q*, and (ii) decreasing *Q* (d*Q*/dt ≤ 0) when *P* does not exceed *Q*. This can be thought as the percentage of days when $S_{dyn}$ is the main control of *Q*, with $S_{dyn}$ replenishing when *P* exceeds *Q* (condition i) and depleting otherwise, with no additional main sources of sustainment for *Q* (conditions ii). We found that all catchments met these conditions for most of the days over the study period (Fig. 3 here). The percentage of days with $S_{dyn}$ as main control of *Q* spanned indeed between 54 and 72% across the catchments. This therefore shows the suitability of the method that we used for the estimation of $S_{dyn}$ for our case study. We propose to add this analysis to a revised version of the manuscript, by rephrasing L144–146 as follows.

Kirchner (2009) showed that $S_{dyn}$ can be derived from *Q* time series and the recession characteristics of the catchments, under the assumption of *Q* mainly controlled by it. To verify this assumption for the study catchments, we followed an approach similar to Kirchner (2009). Specifically, we computed the percentage of winter days (November to April, included) over the study period with (i) rising *Q* (d*Q*/dt > 0) when *P* exceeds *Q* and (ii) decreasing *Q* (d*Q*/dt ≤ 0) otherwise. These conditions indicate limited influence of additional stores which are fed during *P* events and then sustain *Q* during dry periods. The percentages of days for which $S_{dyn}$ can be thought as the main source of *Q* spanned between 54 and 72% across the catchments, meaning that the assumption underlying the method proposed by Kirchner (2009) is met for most of the time in all catchments.

[Figure]

Fig. 3: Distribution of the percentage of days when storage (*S*) is the main control of streamflow (*Q*), across all catchments.

3. *So, how do you quantify the uncertainties in E you derived from water balance as I am not sure the uncertainties in S$_{dyn}$.*

Deriving *E* through a water balance approach may involve considerable uncertainties from the assumptions around potential changes in catchment storage (*S*) and data themselves (see reply to comment #1 by Referee #1). Bruno & Duethmann (2024) extensively studied uncertainties in long-term variations in water balance-derived *E* for small catchments without substantial water management in Germany, by using the same approach and datasets as here. This work revealed that the main source of uncertainty is *P* rather than assumptions regarding *S* and long-term variations in S$_{dyn}$ largely agreed with those from groundwater data. Bruno & Duethmann (2024) furthermore showed that long-term variations in *E* were generally robust in the study region with respect to alternative data sources. We realize that we currently do not discuss potential uncertainties in *E* in light of Bruno & Duethmann (2024). We propose to refer to this work in the Introduction already, by rephrasing L72–73 as follows, and to discuss it in a new section on uncertainties in this study (4.4 Sources of uncertainty).

L72–73: Bruno & Duethmann (2024) reported robust increases in *E* in small catchments in Germany between the 1970s and 2000s, regardless of uncertainties in *P* data and in the consideration of potential *S* changes.

4.4 Sources of uncertainty: Bruno and Duethmann (2024) moreover showed that long-term variations of water balance-derived *E* were generally robust to uncertainties in *P* and *S* for small catchments in Germany over the last five decades, and coherent with those from point-scale *E* data.

4. *Also, I have to say, for the catchments with areas ranging from 50-150km$^2$, the lateral groundwater flow is significant which has been discussed in Ying Fan's paper 'Are catchments leaky?' and also quantified in our research (not published yet). Therefore, Equation 3 might be problematic.*

We agree that estimating *E* from *P* and *Q* observations implies considerable uncertainties for some catchments (Fan, 2019; Kampf et al., 2020; Safeeq et al., 2020). However, here we are mostly interested in long-term variations in *E* at a regional scale. For this, we argue that alternative approaches can be equally challenging, given issues in modelling long-term variations in *E* (Duethmann et al., 2020) and in the representativeness of satellite-derived products for small catchments. Thus, we chose this approach, in line with several previous works (e.g., Teuling et al., 2009; Ukkola et al., 2013; Duethmann and Bloeschl, 2018; Bruno and Duethmann, 2024). Bruno and Duethmann (2024) furthermore compared trends in water balance-derived *E* and in *E* data from lysimeters and flux towers in Central Europe. Despite the general paucity of point-scale data, these showed similar temporal dynamics to water balance-derived *E*. This reinforces the suitability of the water balance-approach to study long-term variations in *E* in our study region. We acknowledge, however, that *E* estimates may be still uncertain for individual catchments and years, due to potential data issues and intercatchment groundwater flows. To tackle these uncertainties, we therefore performed the multiple linear regressions on cluster-, 5-year averages. To provide more context around our methodological choices, the associated uncertainties, and our strategies to minimize them, we plan to rephrase L167–169 and to add more discussion in the new Section 4.4 as follows.

L167–169: We adopted 5-year averages to focus on long-term dynamics and reduce potential uncertainties in water balance-derived *E*. Moreover, for the cluster-scale analysis we used average time series across the catchments in each cluster to minimize uncertainties in *E* for specific catchments, while analysing the main signal at a regional scale.

4.4 Sources of uncertainty: Due to the generally coarse resolution of satellite-derived $E$ estimates, we computed $E$ from observed $P$ and $Q$, and estimates of the $S_{dyn}$ of the catchments from $Q$ data as a first order approximation of $S$. This approach may be problematic for specific catchments, such as those with relevant intercatchment groundwater flows (Fan, 2019; Kampf et al., 2020; Safeeq et al., 2021). Yet, previous works often used a water balance-approach to study long-term variations in $E$ (Teuling et al., 2009; Ukkola and Prentice, 2013; Duethmann and Blöschl, 2018; Bruno and Duethmann 2024). Bruno and Duethmann (2024) moreover showed that long-term variations of water balance-derived $E$ were generally robust to uncertainties in $P$ and $S$ for small catchments in Germany over the last five decades, and coherent with those from point-scale $E$ data. Here, we further aimed at minimizing uncertainties in $E$ for specific catchments and years by using cluster-, 5-year averages in the multiple linear regressions for trend attribution (Section 2.5).

5. *Line 161, $P_{DIF}$ and $P_{MAM}$ are used as proxies of storage processes. How and why they e used as proxies?*

Long-term data on soil moisture and groundwater storage are unavailable for the large number of small catchments that we analyze here. Thus, we used precipitation over winter ($P_{DIF}$) and spring ($P_{MAM}$) as proxies for storage recharge in the seasons preceding the dry period in the study region, as frequently done (Duethmann et al., 2015; Saft et al., 2016; Laaha et al., 2017). Following comment #2 by Referee #1 too, we propose to rephrase L161–162 as follows and to add more discussion on this point in the new Section 4.4.

L161–162: Since long-term data on soil moisture and groundwater storage are not available for the study catchments, we used $P_{MAM}$ and $P_{DJF}$ as proxies of storage recharge in the seasons preceding the dry one (Duethmann et al., 2015; Saft et al., 2016; Laaha et al., 2017).

4.4 Sources of uncertainty: Finally, as potential predictors of changes in summer low flows we approximated storage processes with $P$ in the season preceding the dry one, due to unavailability of long-term $S$ data for the study catchments. We chose this approach instead of using alternative proxies for $S$ (e.g., estimates of $S_{dyn}$ or baseflow from $Q$ data) to avoid dependences between predictors and target variable (summer low flows). The satisfactory performances of the multiple linear regressions and the plausible signs of their coefficients suggest the suitability of the selected predictors to represent the long-term dynamic of summer low flows (Table S1).

6. *All conclusions occur in less than 30% of the catchments, so how do you think about the generality of the study?*

The main findings of this study can be summarized as:

- The magnitude of summer low flows consistently decreased across 363 small catchments with no substantial water management in Germany over 1970–2019 (Fig. 4 in the manuscript);
- Increases in $E$ were a relevant driver of these decreases, especially for catchments in the Eastern area (Fig. 6 and 7);
- Changes in the $P$-$Q$ relationship occurred in catchments with underlying increases in $E$ during a multi-year drought between 1989 and 1993 (Fig. 8).

With respect to first finding, most of the catchments experienced a tendency to decreases in summer low flows, with negative trends in 77% of the catchments and an interquartile range of -7.5/-0.6 % decade[-1] (white boxplot in Fig. 4c). Furthermore, significant negative trends occurred in 31% of the catchments, while significant positive trends in the 2% only (Fig. 4b). To better highlight the general

decreasing tendency in summer low flows across the catchments, we propose to complement L15 and L212–213 as follows.

L15: Summer low flows decreased (increased) significantly in 31 % (2 %) of the catchments, with a median trend of -3.7 % decade$^{-1}$ across all catchments.

L212–213: Trends in 7d$Q_{min, JJA}$ were significantly negative in 31 % of the catchments and significantly positive in 2 % of them (negative in 77 % and positive in 23 %, Fig. 4b), with median (interquartile range, IQR) of -3.7 (-7.5/-0.6) % decade$^{-1}$ across all catchments (Fig. 4c).

Regarding the relevance of increases in $E$ for decreases in summer low flows (second finding) and for changes in the $P$-$Q$ relationship during multi-year droughts (third finding), we acknowledge that additional processes were also important for decreases in summer low flows in specific clusters (Fig. 6 and 7 in the manuscript) and hydrological changes occurred rather sparsely during the multi-year drought under study (see also reply to the comment #5 by Referee #1). However, we believe that our findings are relevant well beyond our study region, given the strong increases in $E$ in many regions worldwide in recent decades (Teuling et al., 2009; Ukkola & Prentice, 2013; Duethmann & Blöschl, 2018; Yang et al., 2023) and their generally limited consideration in low-flow and drought analyses. We plan to highlight better the relevance of our findings outside our study region in a revised version of the manuscript, by adding a paragraph in the Discussion, and rephrasing L396–397 and L410–411 as follows.

Discussion: However, attributing decreases in summer low flows to their causes is still challenging (Montanari et al., 2023) and similarly for decreases in $Q$ generation during multi-year droughts (Fowler et al., 2022). Here, we revealed that increases in $E$ contributed (i) to decreases in summer low flows, in particular in catchments tending to arid, and (ii) to changes in the $P$-$Q$ relationship during the multi-year drought that occurred in Germany between 1989 and 1993. With strong increases in $E$ reported for many regions of the world over recent decades (Teuling et al., 2009; Ukkola & Prentice, 2013; Duethmann & Blöschl, 2018; Yang et al., 2023), these findings are relevant beyond our study region.

L396–397: (…) we underline the importance of monitoring changes in $E$ for the prediction of potential decreases in $Q$ during dry periods, particularly in arid regions.

L410–411: We illustrated the imprint of long-term increases in $E$ on decreases in $Q$ during dry conditions, which can be especially relevant for arid regions and prolonged droughts.

> 7. I am wondering why ET is increasing? The land cover change or the temperature increasing? Authors listed a lot of attributes of the catchments in table 1 but are limited used in the analysis.

In Bruno & Duethmann (2024), we provided a first assessment of potential causes of past changes in $E$ in small catchments in Germany, by showing that these changes were consistent with those in $P$ and solar radiation. However, an in-depth investigation on the causes of past increases in $E$ in the study region, including changes in land cover, would be noteworthy. Yet, we see this analysis as out of the scope of the current study. The catchment attributes in Table 1 are static attributes, which we used for a characterization of the study catchments. Additional, dynamic attributes would be needed to relate changes in $E$ with changes in land cover for instance. To provide some background on possible causes of increases in $E$ to the readers, we suggest adding this point to the Introduction as follows.

Increases in $E$ followed changes in climate and land cover (Duethmann and Blöschl, 2018; Teuling et al., 2019; Yang et al., 2023; Bruno & Duethmann, 2024).

*8. Have you cited the paper talking about the similar thing? Tran et al. (2023), Frontiers in Water.*

Thank you for pointing to this piece of literature which we intend to add to the Introduction and Discussion.

Introduction: Furthermore, increases in *E* contributed to decreases in annual *Q* on the long-term (Teuling et al., 2009; Fischer et al., 2023) and during dry years, especially in mountain catchments (Tran et al., 2023).

Discussion: Our findings on the role of increases in *E* on decreases in summer low flows expand results at annual time scale (Fischer et al., 2023; Tran et al., 2023)

References:

Berghuijs, W. R., Hartmann, A., & Woods, R. A. (2016). Streamflow sensitivity to water storage changes across Europe. *Geophysical Research Letters*, *43*(5), 1980–1987. https://doi.org/10.1002/2016GL067927

Bruno, G., & Duethmann, D. (2024). Increases in Water Balance-Derived Catchment Evapotranspiration in Germany During 1970s–2000s Turning Into Decreases Over the Last Two Decades, Despite Uncertainties. *Geophysical Research Letters*, *51*(6), e2023GL107753. https://doi.org/10.1029/2023GL107753

Court, A. (1962). Measures of streamflow timing. *Journal of Geophysical Research*, *67*(11), 4335–4339. https://doi.org/10.1029/JZ067i011p04335

Duethmann, D., & Blöschl, G. (2018). Why has catchment evaporation increased in the past 40 years? A data-based study in Austria. *Hydrology and Earth System Sciences*, *22*(10), 5143–5158. https://doi.org/10.5194/hess-22-5143-2018

Duethmann, D., Bolch, T., Farinotti, D., Kriegel, D., Vorogushyn, S., Merz, B., Pieczonka, T., Jiang, T., Su, B., & Güntner, A. (2015). Attribution of streamflow trends in snow and glacier melt-dominated catchments of the Tarim River, Central Asia. *Water Resources Research*, *51*(6), 4727–4750. https://doi.org/10.1002/2014WR016716

Fan, Y. (2019). Are catchments leaky? *WIREs Water*, *6*(6), e1386. https://doi.org/10.1002/wat2.1386

Fischer, M., Pavlík, P., Vizina, A., Bernsteinová, J., Parajka, J., Anderson, M., Řehoř, J., Ivančicová, J., Štěpánek, P., Balek, J., Hain, C., Tachecí, P., Hanel, M., Lukeš, P., Bláhová, M., Dlabal, J., Zahradníček, P., Máca, P., Komma, J., … Trnka, M. (2023). Attributing the drivers of runoff decline in the Thaya river basin. *Journal of Hydrology: Regional Studies*, *48*, 101436. https://doi.org/10.1016/j.ejrh.2023.101436

Fowler, K., Peel, M., Saft, M., Peterson, T. J., Western, A., Band, L., Petheram, C., Dharmadi, S., Tan, K. S., Zhang, L., Lane, P., Kiem, A., Marshall, L., Griebel, A., Medlyn, B. E., Ryu, D., Bonotto, G., Wasko, C., Ukkola, A., … Nathan, R. (2022). Explaining changes in rainfall–runoff relationships during and after Australia's Millennium Drought: A community perspective. *Hydrology and Earth System Sciences*, *26*(23), 6073–6120. https://doi.org/10.5194/hess-26-6073-2022

Han, J., Liu, Z., Woods, R., McVicar, T. R., Yang, D., Wang, T., Hou, Y., Guo, Y., Li, C., & Yang, Y. (2024). Streamflow seasonality in a snow-dwindling world. *Nature*, *629*(8014), 1075–1081. https://doi.org/10.1038/s41586-024-07299-y

Kampf, S. K., Burges, S. J., Hammond, J. C., Bhaskar, A., Covino, T. P., Eurich, A., Harrison, H., Lefsky, M., Martin, C., McGrath, D., Puntenney-Desmond, K., & Willi, K. (2020). The Case for an Open Water Balance: Re-envisioning Network Design and Data Analysis for a Complex, Uncertain World. *Water Resources Research*, *56*(6), e2019WR026699. https://doi.org/10.1029/2019WR026699

Kirchner, J. W. (2009). Catchments as simple dynamical systems: Catchment characterization, rainfall-runoff modeling, and doing hydrology backward. *Water Resources Research*, *45*(2). https://doi.org/10.1029/2008WR006912

Laaha, G., Gauster, T., Tallaksen, L. M., Vidal, J.-P., Stahl, K., Prudhomme, C., Heudorfer, B., Vlnas, R., Ionita, M., Van Lanen, H. A. J., Adler, M.-J., Caillouet, L., Delus, C., Fendekova, M., Gailliez, S., Hannaford, J., Kingston, D., Van Loon, A. F., Mediero, L., … Wong, W. K. (2017). The European 2015 drought from a hydrological perspective. *Hydrology and Earth System Sciences*, *21*(6), 3001–3024. https://doi.org/10.5194/hess-21-3001-2017

Montanari, A., Nguyen, H., Rubinetti, S., Ceola, S., Galelli, S., Rubino, A., & Zanchettin, D. (2023). Why the 2022 Po River drought is the worst in the past two centuries. *Science Advances*, *9*(32), eadg8304. https://doi.org/10.1126/sciadv.adg8304

Renner, M., & Hauffe, C. (2024). Impacts of climate and land surface change on catchment evapotranspiration and runoff from 1951 to 2020 in Saxony, Germany. *Hydrology and Earth System Sciences*, *28*(13), 2849–2869. https://doi.org/10.5194/hess-28-2849-2024

Safeeq, M., Bart, R. R., Pelak, N. F., Singh, C. K., Dralle, D. N., Hartsough, P., & Wagenbrenner, J. W. (2021). How realistic are water-balance closure assumptions? A demonstration from the southern sierra critical zone observatory and kings river experimental watersheds. *Hydrological Processes*, *35*(5), e14199. https://doi.org/10.1002/hyp.14199

Saft, M., Peel, M. C., Western, A. W., & Zhang, L. (2016). Predicting shifts in rainfall-runoff partitioning during multiyear drought: Roles of dry period and catchment characteristics. *Water Resources Research*, *52*(12), 9290–9305. https://doi.org/10.1002/2016WR019525

Staudinger, M., Stoelzle, M., Seeger, S., Seibert, J., Weiler, M., & Stahl, K. (2017). Catchment water storage variation with elevation. *Hydrological Processes*, *31*(11), 2000–2015. https://doi.org/10.1002/hyp.11158

Stoelzle, M., Stahl, K., & Weiler, M. (2013). Are streamflow recession characteristics really characteristic? *Hydrology and Earth System Sciences*, *17*(2), 817–828. https://doi.org/10.5194/hess-17-817-2013

Teuling, A. J., De Badts, E. A. G., Jansen, F. A., Fuchs, R., Buitink, J., Hoek Van Dijke, A. J., & Sterling, S. M. (2019). Climate change, reforestation/afforestation, and urbanization impacts on evapotranspiration and streamflow in Europe. *Hydrology and Earth System Sciences*, *23*(9), 3631–3652. https://doi.org/10.5194/hess-23-3631-2019

Teuling, A. J., Hirschi, M., Ohmura, A., Wild, M., Reichstein, M., Ciais, P., Buchmann, N., Ammann, C., Montagnani, L., Richardson, A. D., Wohlfahrt, G., & Seneviratne, S. I. (2009). A regional perspective on trends in continental evaporation. *Geophysical Research Letters*, *36*(2). https://doi.org/10.1029/2008GL036584

Tran, H., Yang, C., Condon, L. E., & Maxwell, R. M. (2023). The Budyko shape parameter as a descriptive index for streamflow loss. *Frontiers in Water*, *5*, 1258367. https://doi.org/10.3389/frwa.2023.1258367

Trotter, L., Saft, M., Peel, M. C., & Fowler, K. J. A. (2024). Recession constants are non-stationary: Impacts of multi-annual drought on catchment recession behaviour and storage dynamics. *Journal of Hydrology*, *630*, 130707. https://doi.org/10.1016/j.jhydrol.2024.130707

Ukkola, A. M., & Prentice, I. C. (2013). A worldwide analysis of trends in water-balance evapotranspiration. *Hydrology and Earth System Sciences*, *17*(10), 4177–4187. https://doi.org/10.5194/hess-17-4177-2013

Yang, Y., Roderick, M. L., Guo, H., Miralles, D. G., Zhang, L., Fatichi, S., Luo, X., Zhang, Y., McVicar, T. R., Tu, Z., Keenan, T. F., Fisher, J. B., Gan, R., Zhang, X., Piao, S., Zhang, B., & Yang, D. (2023). Evapotranspiration on a greening Earth. *Nature Reviews Earth & Environment*, *4*(9), 626–641. https://doi.org/10.1038/s43017-023-00464-3

---

## Author Response (AR1)

Dear Editor,

We thank you and the two Referees for your time in reviewing our manuscript and your constructive feedback on it. Please find below our point-by-point reply to your comments (italic) and the changes we did in the manuscript to address them (underlined). We also attach the revised version of the manuscript, with tracked changes as well.

Best regards,

Giulia Bruno and co-authors

Editor:

*Dear Authors,*

*As suggested by both referees, the manuscript requires major revisions to address the key concerns raised during the review process. These revisions are critical to ensuring that the manuscript meets the expected standards of quality, rigor, and clarity. The feedback provided by the referees includes constructive criticism and detailed recommendations that aim to enhance the overall impact and scientific validity of the work.*

*Additionally, the authors have proposed modifications to further refine the manuscript, incorporating new insights, improved methodologies and adding new figures, tables and citations. These changes will not only respond to the referees' comments but also strengthen the manuscript by addressing the identified gaps and ambiguities.*

*The major revision process shall involve the following steps:*

*• Carefully addressing each point raised by the referees to ensure all concerns are adequately resolved.*
*• Integrating the proposed modifications to improve the structure, clarity, and scientific depth of the manuscript.*

We have now implemented all Referees' suggestions in a revised version of the manuscript. In particular, we introduced additional analyses regarding (i) the multiple linear regressions at the catchment scale as suggested by Referee #1, and (ii) long-term variations in alternative streamflow ($Q$) metrics and (iii) the evaluation of our approach for the estimation of the dynamic storage ($S_{dyn}$) of the catchments as suggested by Referee #2. Also, we discussed in more detail the reasons for our methodological choices and potential uncertainties in the study, the latter in a new section (4.4. Sources of uncertainty).

Referee #1:

*This paper analysed the summer low flows by using the datasets from 363 small catchments. The relationships among evapotranspiration (E), precipitation (P) and streamflow (Q), as well as the storage (S) were quantified. Results showed that summer low flows decreased significantly, and increased E played the main driver in the eastern catchments. In addition, the P-Q relationship changed in 26% of catchments between 1989 and 1993. Generally, the structure of the paper is clear and well-organized, however, there are some concerns.*

We thank you for appreciating the clarity of our manuscript and for raising interesting points to improve our work.

*Specific comments:*

1. *The coefficients of a and b in equation (2) should be vary with wind direction and elevation of gauges. Does authors calculate them for each gauges? If so, please provide the analysis which are the main uncertainty for precipitation datasets.*

The method we used for the correction of precipitation (*P*) for gauge undercatch was developed for point data (Richter, 1995), with coefficients depending on gauge characteristics (wind exposure) and meteorological conditions (*P* type, Eq. 2). Here, we used a gridded *P* dataset (Sect. 2.2) and we applied the correction procedure at pixel-scale, by assuming all pixels as moderately sheltered with respect to wind exposure. This approach was used also in previous works (Duethmann & Blöschl, 2018; Bruno & Duethmann, 2024). Duethmann & Blöschl (2018) showed that alternative assumptions regarding the corrective coefficients had small influences on long-term variations in *P* and water-balance derived catchment evapotranspiration (*E*), we are mostly interested in for our analyses. Thus, we argue that this assumption unlikely affected our main conclusions. We agree, however, that *P* data are unavoidably affected by some degree of uncertainty, either per se and following this correction, which we did not discuss in the original manuscript. We modified L139–143 to make clearer our assumption regarding the correction procedure and we added some discussion concerning the uncertainty in *P* data in the new Sect. 4.4.

L139–143: We corrected the dataset for gauge undercatch following the method proposed for Germany by Richter (1995):

$$\underline{P_{corr} = P_{uncorr} + aP_{uncorr}^{\ b}} \qquad \underline{(2)}$$

with $P_{corr}$ as corrected *P*, $P_{uncorr}$ uncorrected *P*, *a* and *b* coefficients which vary with wind exposure of the gauges, precipitation type (rain or snow), and season. Here, we assumed for all grid cells the coefficients for moderately sheltered locations in Richter (1995), given the low sensitivity of long-term variations in *P* to the selected coefficients (Duethmann & Blöschl, 2018).

Sect. 4.4: Uncertainties in *P* can arise from potential inhomogeneities in gauge data over time and gauge undercatch. Here, we used a gridded *P* dataset from the interpolation of a fixed number of gauges over time to minimize inhomogeneities and we corrected it for gauge undercatch (Sect. 2.2). This correction procedure may lead to further uncertainties, but Duethmann and Blöschl (2018) demonstrated that assumptions regarding corrective coefficients have little influence on trends in *P* and *E*, we were mostly interested in here.

2. *It is quite difficult to understand the equation (4), where the dynamics of storage was approximated by P_mam and P_djf. Please provide more explanations. In addition, since baseflow is 0.66 in this area, which means the soil moisture and groundwater both plays important roles in runoff variation. But it seems they are not taken into account in the analysis.*

We agree that soil and groundwater storage are relevant for *Q* generation in the study catchments. As predictors of trends in summer low flows, we indeed considered variations in *E*, summer *P*, and storage (*S*). We accounted for the influence of *S* variations using winter precipitation ($P_{DIF}$) and spring precipitation ($P_{MAM}$) as wetness conditions in the seasons preceding the summer and therefore, proxies for *S* recharge. We used these proxies since long-term data on soil moisture and groundwater are unavailable for the large sample of small catchments that we analyzed, similarly to what done by previous works (Duethmann et al., 2015; Saft et al., 2016; Laaha et al., 2017). We rephrased L174–176 as follows and discussed this point in the new Sect. 4.4.

L174–176: We used $P_{MAM}$ and $P_{DJF}$ as proxies of storage recharge in the seasons preceding the summer similarly to what done by previous work to overcome the unavailability of long-term data on soil moisture and groundwater storage for the study catchments (Duethmann et al., 2015; Saft et al., 2016; Laaha et al., 2017).

Sect. 4.4: Finally, as potential predictors of changes in summer low flows we approximated storage processes with $P$ in the season preceding summer, due to unavailability of long-term $S$ data for the study catchments. We chose this approach instead of using alternative proxies for $S$ (e.g., estimates of $S_{dyn}$ or baseflow from $Q$ data) to avoid dependences between predictors and target variable (summer low flows). The satisfactory performances of the multiple linear regressions and the plausible signs of their coefficients suggest the suitability of the selected predictors to represent the long-term dynamic of summer low flows (Table S1 and Fig. S6).

3. *For multiple linear regression in predicting summer low flows, the authors showed the R2 in four clusters which exhibited good performance in Table S1. However, there is spatial variation among different gauges so the coefficients vary at each gauge, does the regression in the gauge scale follow the same trend with cluster?*

In a first step, we performed the multiple linear regression at a cluster-scale to minimize uncertainties in $E$ for specific catchments (see also reply to comment #4 by Referee #2). However, we see that such analysis may not fully reveal potential spatial differences in the predictors of the temporal dynamics of summer low flows. Thus, we repeated the analysis at the catchment-scale. We achieved overall satisfactory results in terms of model performances (median coefficient of determination across the catchments equal to 0.78, Fig. 1a and b here) and in line with those at a cluster-scale. By looking at the predictor with highest contribution to the simulation (primary predictor) for each catchment, we found that summer precipitation ($P_{JJA}$) was the most recurrent one across all clusters (Fig. 1c). $P_{JJA}$ was frequently non-significant though, especially where model performances were relatively low (Fig. 1a) and predictors had a similar relative contribution (not shown). By focusing on significant primary predictors only, the most recurrent ones were $P_{JJA}$ in the Pre-Alpine cluster, $P_{MAM}$ in the South-Central one, and $E$ in the Eastern and Northern clusters (Fig. 1d). These are coherent with the conclusions we draw at the cluster-scale (i.e., $P_{JJA}$ dominant predictor in the Pre-Alpine cluster, $P_{MAM}$ significant predictor in the South-Central one, and $E$ in the Eastern and Northern clusters, Fig. 1). To reinforce our trend attribution in the revised version of the manuscript, we added this analysis with Fig. 1 here as new Fig. S6 in the Supplement, and adapted the description of methods and results as follows.

[Figure]

Fig. 1: Model results from the multiple linear regressions on catchment-scale data. (a) Map of coefficients of determination of the models ($R^2$). (b) Histogram of $R^2$. (c) Map of primary predictors (black edges if significant). (d) Relative frequency of primary predictors by cluster. Light grey in (a, c, and d) refers to catchments with high multicollineairty of the predictors, and thus excluded from the analysis (Sect. 2.5). In (d), Pre-al. refers to Pre-Alpine, South. to South-Central, East. to Eastern, and North. to Northern cluster.

Methods, L178–183: Firstly, we used multiple linear regression to model the temporal dynamics of summer low flows ($7dQ_{\text{min, JJA}}$ in Eq. (4)), from the dynamics of the predictors, both at a catchment- and cluster-scale:

$$7dQ_{min, \ JJA} = \alpha_1 E + \alpha_2 P_{JJA} + \alpha_3 P_{MAM} + \alpha_4 P_{DJF} + \varepsilon \qquad (4)$$

with $\alpha_i$ ($i$ = 1…4) the regression coefficient for each predictor and $\varepsilon$ the model residuals. We adopted 5-year averages to focus on long-term dynamics and reduce uncertainties in water balance-derived $E$. Moreover, for the cluster-scale analysis we used average time series across the catchments in each cluster to minimize uncertainties in $E$ for specific catchments, while analysing the main signal at a regional scale.

Results, L256–258: Multiple linear regression for predicting long-term dynamics in summer low flows achieved satisfactory performances both at cluster- and catchment-scale (for $7dQ_{\text{min, JJA}}$, $R^2 > 0.7$ for each cluster and median $R^2$ of 0.78 across all catchments, Table S1, and Fig. S6a and b).

L262–265: Catchment-scale results showed similar patterns, despite being unavoidably more affected by noise than those at the cluster-scale (Fig. S6c and d). By focusing on significant primary predictors only (i.e., predictors with highest contribution to the simulations), $P_{\text{JJA}}$ was the most recurrent one in the Pre-Alpine cluster, $P_{\text{MAM}}$ in the South-Central cluster, and $E$ in the Eastern and Northern clusters (Fig. S6d).

*4. Could authors discuss why the r= -0.49 in P_djf for the easter cluster in lines 258?*

In the Eastern cluster, we indeed detected a negative correlation between trends in summer low flows and trends in $P_{\text{DJF}}$, which may sound counterintuitive under our assumption of $P_{\text{DJF}}$ as a proxy for storage recharge. While we acknowledge that spurious effects in the correlation analysis may play a role here, we see a mechanistic explanation, related to $E$-storage feedbacks (Boeing et al., 2024) and the observed changes in this cluster. In particular, catchments in the Eastern cluster showed both positive and negative trends in $P_{\text{DJF}}$, but generally negative trends in $P_{\text{MAM}}$ and positive trends in $E$. This means that increases in $P_{\text{DJF}}$, and thus in storage conditions at the beginning of the growing season, might have buffered the decreases in $P_{\text{MAM}}$ in sustaining the increases in $E$ in some catchments. Increases in $E$, in turn, contributed to decreases in summer low flows in this cluster (Fig. 6 and 7). Thus, increases in $P_{\text{DJF}}$ may have indirectly contributed to decreases in summer low flows in the Eastern cluster. In the revised manuscript, we expanded the mechanistic explanation on this point in the Discussion as follows.

L347–349: These three mechanisms (i.e., widespread increases in $E$, local decreases in $P_{\text{JJA}}$, and local decreases in $P_{\text{MAM}}$, in the previous sentence not reported here) also overcompensated local increases in storage recharge during winter (approximated by $P_{\text{DJF}}$), possibly through storage depletion by $E$ (Boeing et al., 2024) for instance in the Eastern cluster (Fig. 7).

*5. For 363 catchments, there is only 15 catchment with negative Cp-q rel values which distributed sparsely in Fig.8(a). I am curious about the possibility caused by data process uncertainty.*

Having no changes in the *P-Q* relationships even during prolonged dry periods was common expectation for humid catchments until recently (Massari et al., 2022) and a sparse occurrence of these changes is positive for water management (see e.g. Fowler et al., 2022 for practical implications of these changes). We see that our study revealed less widespread changes in the *P-Q* relationship during the multi-year drought in Germany in the early 1990s than previous works for other case studies. We discuss that these differences may relate to the characteristics of the multi-year droughts (i.e., severity and duration of the *P* deficits) and to the hydro-climatic properties of the catchments. We agree that the unavoidable uncertainty in *P* and *Q* data (see reply to comment #1) may also play a role, despite we here aimed at minimizing uncertainty in *P* data by using a dataset specifically tailored to long-term consistency (Sect. 2.2). To add this point to the Discussion, we rephrased L390–395 as follows.

L390–395: According to this analysis, the multi-year drought in Germany in the early 1990s had less severe impact on *Q* generation than the Millennium drought in Australia (changes in 56 % of the catchments, with median decrease of approximately -50 %, Saft et al. 2015), the 2012–2016 event in California (mean decreases of -28 % across three catchments, Avanzi et al. 2020), and the 2010–2020 drought in Chile (changes in 61 % of the catchments and mean decrease of -19 %, Alvarez-Garreton et al., 2021). These differences may be related to the characteristics of the *P* anomalies (severity and duration), potential uncertainties in the underlying data (Sect. 4.4), and the hydro-climatic characteristics of the catchments.

6. *For Fig.5 and Table S1, it showed that P_jja played a major contribution to Q changes in Pre-Alpine and South-Central cluster, while E played much more contribution in Eastern and Northern area. Does it relate to elevation changes?*

Catchments in the Pre-Alpine and South-Central clusters have indeed relatively higher mean elevations than others (Fig. 3 in the manuscript). Furthermore, catchments in the Pre-Alpine cluster generally experienced decreases in $P_{JJA}$ and mild increases in *E*. Catchments in the South-Central cluster showed similar behaviors, but overall small trends. On the contrary, catchments in the Eastern and Northern clusters largely had increases in $P_{JJA}$ and *E*. Therefore, we argue that differences in the main drivers of decreases in summer low flows between the Pre-Alpine and North-Eastern areas can be ascribed to differences in the variations in the drivers themselves, rather than to differences in catchment characteristics like elevation. To make the differences in hydro-climatic changes among the clusters easier to grasp, we added the following summary table in Sect. 4.2.

Table 1: Summary of long-term variations in summer low flows and in their potential predictors (annual evapotranspiration, *E*, and precipitation over summer, $P_{JJA}$, spring, $P_{MAM}$, and winter, $P_{DJF}$) over 1970–2019, and drivers of variations in summer low flows (only significant ones according to both attribution analyses, Sect. 2.5) for each cluster. Red arrows refer to strong decreases, light red arrows to mild decreases, light blue arrows to mild increases, and blue arrows to strong increases at the median level, with ± 2 % decade$^{-1}$ as thresholds for strong increases/decreases.

| | Cluster | | | |
|---|---|---|---|---|
| | Pre-Alpine | South-Central | Eastern | Northern |
| Variations in summer low flows | ↓ (light red) | ↓ (red) | ↓ (red) | ↓ (red) |
| Variations in *E* | ↑ (light blue) | ↑ (light blue) | ↑ (blue) | ↑ (blue) |
| Variations in $P_{JJA}$ | ↓ (light red) | ↓ (light red) | ↑ (blue) | ↑ (light blue) |
| Variations in $P_{MAM}$ | ↑ (light blue) | ↓ (light red) | ↓ (light red) | ↓ (red) |
| Variations in $P_{DJF}$ | ↑ (blue) | ↑ (blue) | ↑ (blue) | ↑ (blue) |
| Drivers of variations in summer low flows | $P_{JJA}$ | - | *E* | *E*, $P_{JJA}$ |

Referee #2:

*Bruno et al. analyzed the decreasing trend of low flow in 363 small catchments in Germany. They also attributed the decrease to the increase of ET. They further unraveled that the change of P-Q relationship during drought produces lower flow which is generally due to the increased ET. They conducted this work based on observations of P and Q, empirical expression of subsurface storage, and ET derived based on water balance and statistical analysis. I think studying the decrease of low flow and trying to find the major drivers is very important in the climate change background. The data and analysis are generally reliable, the structure and the writing are good. However, there the following concerns which need clarification from authors for further review.*

We thank you for your overall positive evaluation of our manuscript and for raising interesting points to improve our work.

1. *I was more or less confused by the overall idea of the authors. If you wanted to check if the decrease of the flow is caused by increased ET, and you also have calculated the water balance, why not do a straightforward analysis of the overall change of P, ET, Q, and S. Then it is easy to get if the decrease of flow is mainly driven by ET. Then you can do the analysis in your manuscript as a follow-up. Otherwise, I feel the conclusions are even not that convincing as, for example, the decrease of low flow might be just because of the shift of the timing of streamflow.*

We understand that you suggest evaluating here the impact of changes in $E$ on $Q$ at an annual time scale. This has already been well documented for several regions and periods, partly overlapping with our case study as well (Teuling et al., 2009; Fischer et al., 2023; Renner & Hauffe, 2024). Here, we rather aimed at assessing the impact of increases in $E$ on $Q$ during dry periods specifically, to complement these previous findings. We see, however, that assessing changes in additional $Q$ metrics, such as annual $Q$ and its timing, may provide useful context regarding hydrological changes in the region and the decreases in the magnitude of summer low flows that we observed. Therefore, we quantified changes in annual $Q$ and its timing, the latter through the center of mass date of $Q$ ($CMD_Q$, Court, 1962; Han et al., 2024). Annual $Q$ and $CMD_Q$ generally showed decreases across the catchments between 1970 and 2019 (Fig. 2g-l here). However, these decreases were less significant than those in the magnitude of summer low flows, with a median trend in annual $Q$ ($CMD_Q$) of -1.3 % decade$^{-1}$ (-0.006 month year$^{-1}$) across all catchments and significant negative trends in 10 (3) % of them (Fig. 2g-l here), as compared to a median trend in $7dQ_{min, JJA}$ of -3.7 % decade$^{-1}$ across all catchments and significant negative trends in 31 % (Fig. 4).

[Figure]

Fig. 2: Long-term variations in streamflow ($Q$) metrics over 1970–2019 (30d$Q_{min, JJA}$, panels a–c, and 7d$Q_{min, M-O}$, panels d–f, as metrics for summer low flows, annual $Q$, panels g–i, and the center of mass date of streamflow CMD$_Q$, panels j–l). (a, d, g, and j) Average anomalies across the study catchments. (b, e, h, and k) Map of catchment-scale trends (black edges if significant). (c, f, i, and l) Boxplots of trends for all catchments and by cluster.

Therefore, summer low flows in particular decreased across the study catchments between 1970 and 2019, as compared to other $Q$ metrics, and these decreases did not occur simply with an overall decrease in $Q$ or a shift in its timing. We introduced this analysis in the revised version of the manuscript, by adding Fig. 2 here as new Fig. S4 in the Supplement, and its methodological aspects, results, and discussion in the corresponding sections.

Methods, L166–168: We also characterised general long-term variations in $Q$, and considered annual $Q$ and the centre of mass date of $Q$ (CMD$_Q$). CMD$_Q$ is the time of the year when half of the annual $Q$ has occured and as such, a metric for $Q$ timing (Court, 1962; Han et al., 2024).

Results, L230–233: Additional $Q$ metrics similarly decreased over 1970–2019, even though less significantly than summer low flows (median trend in annual $Q$ of -1.3 % decade[-1] across all the catchments and significant negative trends in 10 % of them, Fig. S4g-i, and median trend in CMD$_Q$ of -0.006 month year[-1] across all catchments and significant negative trends in 3 % of them, Fig. S4j-l).

Discussion, L325–328: Furthermore, decreases in annual $Q$ and in its timing (CMD$_Q$) were more elusive than those in summer low flows (Fig. S4g-l), meaning that summer low flows in particular decreased

across the study catchments over 1970–2019, and likely not simply due to a general decrease in $Q$ or a shift in its timing.

Furthermore, we added more context around current knowledge on the effect of increases in $E$ on decreases in $Q$ at an annual time scale in the Introduction and Discussion as follows, also to highlight better the novelty of our study with respect to previous literature.

Introduction, L55–57: These increases in $E$ contributed to decreases in annual $Q$ on the long-term (Teuling et al., 2009; Fischer et al., 2023; Renner & Hauffe, 2024) and during dry years, especially in mountain catchments (Tran et al., 2023).

Discussion, L359–361: Our findings on the role of increases in $E$ on decreases in summer low flows expand results at annual time scale (Fischer et al., 2023; Tran et al., 2023), and those by Montanari et al. (2023) on concomitant decreases in summer low flows and increases in $E$ in northern Italy over recent decades.

2.  *You mentioned you used Kirchner's approach to calculate $S_{dyn}$ which needs that S is the main control of Q generation. I am wondering if the 363 catchments you used meet this requirement and where is your analysis for this?*

We agree that the assumption of $Q$ mainly controlled by $S_{dyn}$ may not always hold true across large samples of catchments, despite previous studies adopted it (Staudinger et al., 2017; Trotter et al., 2024) also in our study region (Stoelzle et al., 2013; Berghuijs et al., 2016). We have now evaluated this assumption similarly to what done by Kirchner (2009). Specifically, we computed the percentage of winter days (November to April, included) with (i) rising $Q$ ($dQ/dt > 0$) when precipitation ($P$) exceeds $Q$, and (ii) decreasing $Q$ ($dQ/dt \leq 0$) when $P$ does not exceed $Q$. This can be thought as the percentage of days when $S_{dyn}$ is the main control of $Q$, with $S_{dyn}$ replenishing when $P$ exceeds $Q$ (condition i) and depleting otherwise, with no additional main sources of sustainment for $Q$ (conditions ii). We found that all catchments met these conditions for most of the days over the study period (Fig. 3 here). The percentage of days with $S_{dyn}$ as main control of $Q$ spanned indeed between 54 and 72 % across the catchments. This therefore shows the suitability of the method that we used for the estimation of $S_{dyn}$ for our case study. We added this analysis to the revised version of the manuscript, by rephrasing L150–157 as follows.

Kirchner (2009) showed that $S_{dyn}$ can be derived from the $Q$ time series and recession characteristics of the catchments, under the assumption of $Q$ mainly controlled by it. To verify this assumption for the study catchments, we followed an approach similar to Kirchner (2009). Specifically, we computed the percentage of winter days (November to April, included) over the study period with (i) rising $Q$ ($dQ/dt > 0$) when $P$ exceeds $Q$ and (ii) decreasing $Q$ ($dQ/dt \leq 0$) otherwise. These conditions indicate limited influence of additional stores which are fed during $P$ events and then sustain $Q$ during dry periods. The percentages of days for which $S_{dyn}$ can be thought as the main source of $Q$ spanned between 54 and 72% across the catchments, meaning that the assumption underlying the method proposed by Kirchner (2009) is met for most of the time in all catchments.

[Figure]

Fig. 3: Distribution of the percentage of days when storage ($S$) is the main control of streamflow ($Q$), across all catchments.

3. *So, how do you quantify the uncertainties in E you derived from water balance as I am not sure the uncertainties in $S_{dyn}$.*

Deriving $E$ through a water balance approach may involve considerable uncertainties from the assumptions around potential changes in $S$ and data themselves (see reply to comment #1 by Referee #1). Bruno & Duethmann (2024) extensively studied uncertainties in long-term variations in water balance-derived $E$ for small catchments without substantial water management in Germany, by using the same approach and datasets as here. This work revealed that the main source of uncertainty is $P$ rather than assumptions regarding $S$ and long-term variations in $S_{dyn}$ largely agreed with those from groundwater data. Bruno & Duethmann (2024) furthermore showed that long-term variations in $E$ were generally robust in the study region with respect to alternative data sources. We realized that we did not discuss potential uncertainties in $E$ in light of Bruno & Duethmann (2024) in the original version of the manuscript. In the revised version, we referred to this work in the Introduction already, by rephrasing L76–77 as follows, and discussed it in the new Sect. 4.4.

L76–77: Bruno & Duethmann (2024) reported robust increases in water balance-derived $E$ in small catchments in Germany between the 1970s and 2000s, regardless of uncertainties in $P$ data and in the consideration of potential $S$ changes.

Sect. 4.4: Bruno and Duethmann (2024) moreover showed that long-term variations of water balance-derived $E$ were generally robust to uncertainties in $P$ and $S$ for small catchments in Germany over the last five decades, and coherent with those from point-scale $E$ data.

4. *Also, I have to say, for the catchments with areas ranging from 50-150km$^2$, the lateral groundwater flow is significant which has been discussed in Ying Fan's paper 'Are catchments leaky?' and also quantified in our research (not published yet). Therefore, Equation 3 might be problematic.*

We agree that estimating $E$ from $P$ and $Q$ observations implies considerable uncertainties for some catchments (Fan, 2019; Kampf et al., 2020; Safeeq et al., 2020). However, here we are mostly interested in long-term variations in $E$ at a regional scale. For this, we argue that alternative approaches can be equally challenging, given issues in modelling long-term variations in $E$ (Duethmann et al., 2020) and in the representativeness of satellite-derived products for small catchments. Thus, we chose this approach, in line with several previous works (e.g., Teuling et al., 2009; Ukkola et al.,

2013; Duethmann and Bloeschl, 2018; Bruno and Duethmann, 2024). Bruno and Duethmann (2024) furthermore compared trends in water balance-derived $E$ and in $E$ data from lysimeters and flux towers in Central Europe. Despite the general paucity of point-scale data, these showed similar temporal dynamics to water balance-derived $E$. This reinforces the suitability of the water balance-approach to study long-term variations in $E$ in our study region. We acknowledge, however, that $E$ estimates may be still uncertain for individual catchments and years, due to potential data issues and intercatchment groundwater flows. To tackle these uncertainties, we therefore performed the multiple linear regressions on cluster-, 5-year averages. To provide more context around our methodological choices, the associated uncertainties, and our strategies to minimize them, we rephrased L180–183 and added more discussion in the new Sect. 4.4 as follows.

L180–183: We adopted 5-year averages to focus on long-term dynamics and reduce potential uncertainties in water balance-derived $E$. Moreover, for the cluster-scale analysis we used average time series across the catchments in each cluster to minimize uncertainties in $E$ for specific catchments, while analysing the main signal at a regional scale.

Sect. 4.4: Due to the generally coarse resolution of satellite-derived $E$ estimates, we computed $E$ from observed $P$ and $Q$, and estimates of the $S_{dyn}$ of the catchments from $Q$ data as a first order approximation of $S$. This approach may be problematic for specific catchments, such as those with relevant intercatchment groundwater flows (Fan, 2019; Kampf et al., 2020; Safeeq et al., 2021). Yet, previous works often used a water balance-approach to study long-term variations in $E$ (Teuling et al., 2009; Ukkola and Prentice, 2013; Duethmann and Blöschl, 2018; Bruno and Duethmann 2024). Bruno and Duethmann (2024) moreover showed that long-term variations of water balance-derived $E$ were generally robust to uncertainties in $P$ and $S$ for small catchments in Germany over the last five decades, and coherent with those from point-scale $E$ data. Here, we further aimed at minimizing uncertainties in $E$ for specific catchments and years by using cluster-, 5-year averages in the multiple linear regressions for trend attribution (Sect. 2.5).

5. Line 161, $P_{DIF}$ and $P_{MAM}$ are used as proxies of storage processes. How and why they e used as proxies?

Long-term data on soil moisture and groundwater storage are unavailable for the large number of small catchments that we analyze here. Thus, we used $P_{DIF}$ and $P_{MAM}$ as proxies for storage recharge in the seasons preceding the summer, as frequently done (Duethmann et al., 2015; Saft et al., 2016; Laaha et al., 2017). Following comment #2 by Referee #1 too, we rephrased L174–176 as follows and added more discussion on this point in the new Sect. 4.4.

L174–176: We used $P_{MAM}$ and $P_{DJF}$ as proxies of storage recharge in the seasons preceding the summer similarly to what done by previous work to overcome the unavailability of long-term data on soil moisture and groundwater storage for the study catchments (Duethmann et al., 2015; Saft et al., 2016; Laaha et al., 2017).

Sect. 4.4: Finally, as potential predictors of changes in summer low flows we approximated storage processes with $P$ in the season preceding the summer, due to unavailability of long-term $S$ data for the study catchments. We chose this approach instead of using alternative proxies for $S$ (e.g., estimates of $S_{dyn}$ or baseflow from $Q$ data) to avoid dependences between predictors and target variable (summer low flows). The satisfactory performances of the multiple linear regressions and the

plausible signs of their coefficients suggest the suitability of the selected predictors to represent the long-term dynamic of summer low flows (Table S1 and Fig. S6).

6.  *All conclusions occur in less than 30% of the catchments, so how do you think about the generality of the study?*

The main findings of this study can be summarized as:

- The magnitude of summer low flows consistently decreased across 363 small catchments with no substantial water management in Germany over 1970–2019 (Fig. 4);
- Increases in $E$ were a relevant driver of these decreases, especially for catchments in the Eastern area (Fig. 6 and 7);
- Changes in the *P-Q* relationship occurred in catchments with underlying increases in $E$ during a multi-year drought between 1989 and 1993 (Fig. 8).

With respect to first finding, most of the catchments experienced a tendency to decreases in summer low flows, with negative trends in 77 % of the catchments and an interquartile range of -7.5/-0.6 % decade$^{-1}$ (white boxplot in Fig. 4c). Furthermore, significant negative trends occurred in 31 % of the catchments, while significant positive trends in the 2 % only (Fig. 4b). To better highlight the general decreasing tendency in summer low flows across the catchments, we complemented L20–21 and L225–227 as follows.

L20–21: Summer low flows decreased (increased) significantly in 31 % (2 %) of the catchments, with a median trend of -3.7 % decade$^{-1}$ across all catchments.

L225–227: Trends in $7dQ_{min, JJA}$ were significantly negative in 31 % of the catchments and significantly positive in 2 % of them (negative in 77 % and positive in 23 %, Fig. 4b), with median (interquartile range, IQR) of -3.7 (-7.5/-0.6) % decade$^{-1}$ across all catchments (Fig. 4c).

Regarding the relevance of increases in $E$ for decreases in summer low flows (second finding) and for changes in the *P-Q* relationship during multi-year droughts (third finding), we acknowledge that additional processes were also important for decreases in summer low flows in specific clusters (Fig. 6 and 7) and hydrological changes occurred rather sparsely during the multi-year drought under study (see also reply to the comment #5 by Referee #1). However, we believe that our findings are relevant well beyond our study region, given the strong increases in $E$ in many regions worldwide over recent decades (Duethmann and Blöschl, 2018; Teuling et al., 2019; Yang et al., 2023), and their generally limited consideration in low-flow and drought analyses. We highlighted better the relevance of our findings outside our study region in the revised version of the manuscript, by adding a paragraph in the Discussion, and rephrasing L456–457 and L470–471 as follows.

Discussion, L449–454: However, challenges remain in understanding the causes of decreases in summer low flows (Montanari et al., 2023) and in $Q$ generation during multi-year droughts (Fowler, Peel, Saft, Peterson, et al., 2022). Here, we revealed that increases in $E$ contributed (i) to decreases in summer low flows, in particular in catchments tending to arid, and (ii) to changes in the *P-Q* relationship during the multi-year drought that occurred in Germany between 1989 and 1993. With strong increases in $E$ reported for many regions of the world over recent decades (Duethmann and Blöschl, 2018; Teuling et al., 2019; Yang et al., 2023), these findings are relevant beyond our study region.

L456–457: (…) we underline the importance of monitoring changes in $E$ for the prediction of potential decreases in $Q$ during dry periods, particularly in arid regions.

L470–471: We illustrated the imprint of long-term increases in *E* on decreases in *Q* during dry conditions, which can be especially relevant for arid regions and prolonged droughts.

> 7. *I am wondering why ET is increasing? The land cover change or the temperature increasing? Authors listed a lot of attributes of the catchments in table 1 but are limited used in the analysis.*

In Bruno & Duethmann (2024), we provided a first assessment of potential causes of past changes in *E* in small catchments in Germany, by showing that these changes were consistent with those in *P* and solar radiation. However, an in-depth investigation on the causes of past increases in *E* in the study region, including changes in land cover, would be noteworthy. Yet, we see this analysis as out of the scope of the current study. The catchment attributes in Table 1 are static attributes, which we used for a characterization of the study catchments. Additional, dynamic attributes would be needed to relate changes in *E* with changes in land cover for instance. To provide some background on possible causes of increases in *E* to the readers, we added this point to the Introduction as follows.

L54–55: Over recent decades, *E* has increased in many regions following changes in climate and land cover (Duethmann and Blöschl, 2018; Teuling et al., 2019; Yang et al., 2023; Bruno & Duethmann, 2024).

> 8. *Have you cited the paper talking about the similar thing? Tran et al. (2023), Frontiers in Water.*

Thank you for pointing to this piece of literature which we added to the Introduction and Discussion.

L56–57: These increases in *E* contributed to decreases in annual *Q* on the long-term (Teuling et al., 2009; Fischer et al., 2023) and during dry years, especially in mountain catchments (Tran et al., 2023).

L359–360: Our findings on the role of increases in *E* on decreases in summer low flows expand results at annual time scale (Fischer et al., 2023; Tran et al., 2023)

References:

Alvarez-Garreton, C., Boisier, J. P., Garreaud, R., Seibert, J., & Vis, M. (2021). Progressive water deficits during multiyear droughts in basins with long hydrological memory in Chile. *Hydrology and Earth System Sciences*, *25*(1), 429–446. https://doi.org/10.5194/hess-25-429-2021

Avanzi, F., Rungee, J., Maurer, T., Bales, R., Ma, Q., Glaser, S., & Conklin, M. (2020). Climate elasticity of evapotranspiration shifts the water balance of Mediterranean climates during multi-year droughts. *Hydrology and Earth System Sciences*, *24*(9), 4317–4337. https://doi.org/10.5194/hess-24-4317-2020

Berghuijs, W. R., Hartmann, A., & Woods, R. A. (2016). Streamflow sensitivity to water storage changes across Europe. *Geophysical Research Letters*, *43*(5), 1980–1987. https://doi.org/10.1002/2016GL067927

Boeing, F., Wagener, T., Marx, A., Rakovec, O., Kumar, R., Samaniego, L., & Attinger, S. (2024). Increasing influence of evapotranspiration on prolonged water storage recovery in Germany. *Environmental Research Letters*, *19*(2), 024047. https://doi.org/10.1088/1748-9326/ad24ce

Bruno, G., & Duethmann, D. (2024). Increases in Water Balance-Derived Catchment Evapotranspiration in Germany During 1970s–2000s Turning Into Decreases Over the Last Two Decades, Despite Uncertainties. *Geophysical Research Letters*, *51*(6), e2023GL107753. https://doi.org/10.1029/2023GL107753

Court, A. (1962). Measures of streamflow timing. *Journal of Geophysical Research*, *67*(11), 4335–4339. https://doi.org/10.1029/JZ067i011p04335

Duethmann, D., & Blöschl, G. (2018). Why has catchment evaporation increased in the past 40 years? A data-based study in Austria. *Hydrology and Earth System Sciences*, *22*(10), 5143–5158. https://doi.org/10.5194/hess-22-5143-2018

Duethmann, D., Blöschl, G., & Parajka, J. (2020). Why does a conceptual hydrological model fail to correctly predict discharge changes in response to climate change? *Hydrology and Earth System Sciences*, *24*(7), Article 7. https://doi.org/10.5194/hess-24-3493-2020

Duethmann, D., Bolch, T., Farinotti, D., Kriegel, D., Vorogushyn, S., Merz, B., Pieczonka, T., Jiang, T., Su, B., & Güntner, A. (2015). Attribution of streamflow trends in snow and glacier melt-dominated catchments of the T arim R iver, Central A sia. *Water Resources Research*, *51*(6), 4727–4750. https://doi.org/10.1002/2014WR016716

Fan, Y. (2019). Are catchments leaky? *WIREs Water*, *6*(6), e1386. https://doi.org/10.1002/wat2.1386

Fischer, M., Pavlík, P., Vizina, A., Bernsteinová, J., Parajka, J., Anderson, M., Řehoř, J., Ivančicová, J., Štěpánek, P., Balek, J., Hain, C., Tachecí, P., Hanel, M., Lukeš, P., Bláhová, M., Dlabal, J., Zahradníček, P., Máca, P., Komma, J., … Trnka, M. (2023). Attributing the drivers of runoff decline in the Thaya river basin. *Journal of Hydrology: Regional Studies*, *48*, 101436. https://doi.org/10.1016/j.ejrh.2023.101436

Fowler, K., Peel, M., Saft, M., Nathan, R., Horne, A., Wilby, R., McCutcheon, C., & Peterson, T. (2022). Hydrological Shifts Threaten Water Resources. *Water Resources Research*, *58*(8), Article 8. https://doi.org/10.1029/2021WR031210

Fowler, K., Peel, M., Saft, M., Peterson, T. J., Western, A., Band, L., Petheram, C., Dharmadi, S., Tan, K. S., Zhang, L., Lane, P., Kiem, A., Marshall, L., Griebel, A., Medlyn, B. E., Ryu, D., Bonotto, G., Wasko, C., Ukkola, A., … Nathan, R. (2022). Explaining changes in rainfall–runoff relationships during and after Australia's Millennium Drought: A community perspective. *Hydrology and Earth System Sciences*, *26*(23), 6073–6120. https://doi.org/10.5194/hess-26-6073-2022

Han, J., Liu, Z., Woods, R., McVicar, T. R., Yang, D., Wang, T., Hou, Y., Guo, Y., Li, C., & Yang, Y. (2024). Streamflow seasonality in a snow-dwindling world. *Nature*, *629*(8014), 1075–1081. https://doi.org/10.1038/s41586-024-07299-y

Kampf, S. K., Burges, S. J., Hammond, J. C., Bhaskar, A., Covino, T. P., Eurich, A., Harrison, H., Lefsky, M., Martin, C., McGrath, D., Puntenney-Desmond, K., & Willi, K. (2020). The Case for an Open Water Balance: Re-envisioning Network Design and Data Analysis for a Complex, Uncertain World. *Water Resources Research*, *56*(6), e2019WR026699. https://doi.org/10.1029/2019WR026699

Kirchner, J. W. (2009). Catchments as simple dynamical systems: Catchment characterization, rainfall-runoff modeling, and doing hydrology backward: CATCHMENTS AS SIMPLE DYNAMICAL SYSTEMS. *Water Resources Research*, *45*(2). https://doi.org/10.1029/2008WR006912

Laaha, G., Gauster, T., Tallaksen, L. M., Vidal, J.-P., Stahl, K., Prudhomme, C., Heudorfer, B., Vlnas, R., Ionita, M., Van Lanen, H. A. J., Adler, M.-J., Caillouet, L., Delus, C., Fendekova, M., Gailliez, S., Hannaford, J., Kingston, D., Van Loon, A. F., Mediero, L., … Wong, W. K. (2017). The European 2015 drought from a hydrological perspective. *Hydrology and Earth System Sciences*, *21*(6), 3001–3024. https://doi.org/10.5194/hess-21-3001-2017

Massari, C., Avanzi, F., Bruno, G., Gabellani, S., Penna, D., & Camici, S. (2022). Evaporation enhancement drives the European water-budget deficit during multi-year droughts. *Hydrology and Earth System Sciences*, *26*(6), 1527–1543. https://doi.org/10.5194/hess-26-1527-2022

Montanari, A., Nguyen, H., Rubinetti, S., Ceola, S., Galelli, S., Rubino, A., & Zanchettin, D. (2023). Why the 2022 Po River drought is the worst in the past two centuries. *Science Advances*, *9*(32), eadg8304. https://doi.org/10.1126/sciadv.adg8304

Renner, M., & Hauffe, C. (2024). Impacts of climate and land surface change on catchment evapotranspiration and runoff from 1951 to 2020 in Saxony, Germany. *Hydrology and Earth System Sciences*, *28*(13), 2849–2869. https://doi.org/10.5194/hess-28-2849-2024

Richter, D. (1995). *Ergebnisse methodischer Untersuchungen zur Korrektur des systematischen Messfehlers des Hellmann- Niederschlagsmessers. Offenbach.*

Safeeq, M., Bart, R. R., Pelak, N. F., Singh, C. K., Dralle, D. N., Hartsough, P., & Wagenbrenner, J. W. (2021). How realistic are water-balance closure assumptions? A demonstration from the southern sierra critical zone observatory and kings river experimental watersheds. *Hydrological Processes*, *35*(5), e14199. https://doi.org/10.1002/hyp.14199

Saft, M., Peel, M. C., Western, A. W., & Zhang, L. (2016). Predicting shifts in rainfall-runoff partitioning during multiyear drought: Roles of dry period and catchment characteristics. *Water Resources Research*, *52*(12), 9290–9305. https://doi.org/10.1002/2016WR019525

Saft, M., Western, A. W., Zhang, L., Peel, M. C., & Potter, N. J. (2015). The influence of multiyear drought on the annual rainfall-runoff relationship: An A ustralian perspective. *Water Resources Research*, *51*(4), 2444–2463. https://doi.org/10.1002/2014WR015348

Staudinger, M., Stoelzle, M., Seeger, S., Seibert, J., Weiler, M., & Stahl, K. (2017). Catchment water storage variation with elevation. *Hydrological Processes*, *31*(11), 2000–2015. https://doi.org/10.1002/hyp.11158

Stoelzle, M., Stahl, K., & Weiler, M. (2013). Are streamflow recession characteristics really characteristic? *Hydrology and Earth System Sciences*, *17*(2), 817–828. https://doi.org/10.5194/hess-17-817-2013

Teuling, A. J., De Badts, E. A. G., Jansen, F. A., Fuchs, R., Buitink, J., Hoek Van Dijke, A. J., & Sterling, S. M. (2019). Climate change, reforestation/afforestation, and urbanization impacts on evapotranspiration and streamflow in Europe. *Hydrology and Earth System Sciences*, *23*(9), 3631–3652. https://doi.org/10.5194/hess-23-3631-2019

Teuling, A. J., Hirschi, M., Ohmura, A., Wild, M., Reichstein, M., Ciais, P., Buchmann, N., Ammann, C., Montagnani, L., Richardson, A. D., Wohlfahrt, G., & Seneviratne, S. I. (2009). A regional perspective on trends in continental evaporation: EVAPORATION TRENDS. *Geophysical Research Letters*, *36*(2), n/a-n/a. https://doi.org/10.1029/2008GL036584

Tran, H., Yang, C., Condon, L. E., & Maxwell, R. M. (2023). The Budyko shape parameter as a descriptive index for streamflow loss. *Frontiers in Water*, *5*, 1258367. https://doi.org/10.3389/frwa.2023.1258367

Trotter, L., Saft, M., Peel, M. C., & Fowler, K. J. A. (2024). Recession constants are non-stationary: Impacts of multi-annual drought on catchment recession behaviour and storage dynamics. *Journal of Hydrology*, *630*, 130707. https://doi.org/10.1016/j.jhydrol.2024.130707

Ukkola, A. M., & Prentice, I. C. (2013). A worldwide analysis of trends in water-balance evapotranspiration. *Hydrology and Earth System Sciences*, *17*(10), 4177–4187. https://doi.org/10.5194/hess-17-4177-2013

Yang, Y., Roderick, M. L., Guo, H., Miralles, D. G., Zhang, L., Fatichi, S., Luo, X., Zhang, Y., McVicar, T. R., Tu, Z., Keenan, T. F., Fisher, J. B., Gan, R., Zhang, X., Piao, S., Zhang, B., & Yang, D. (2023). Evapotranspiration on a greening Earth. *Nature Reviews Earth & Environment*, *4*(9), 626–641. https://doi.org/10.1038/s43017-023-00464-3

---

## Author Response (AR2)

Dear Editor,

We thank you for your time in reviewing our revised manuscript and your further comments on it. We have now implemented all your suggestions to improve the clarity of the manuscript. Furthermore, we have analysed the influence of the correction procedure for precipitation, in order to rely less on previous studies and provide evidence from our data. Please find more details below in our point-by-point replies to your comments (italic) and the changes we did in the manuscript to address them (underlined). We further attach a revised version of the manuscript, with tracked changes as well.

Best regards,

Giulia Bruno and co-authors

*Technical Comments and Clarification for Improvement:*

*1- Clarification on PDJF and Summer Low Flows (Referee #1, Question 4):*

*In the author's response to Referee #1 regarding Question 4, further clarification is needed on how increases in PDJF (precipitation during December–February) could have indirectly contributed to decreases in summer low flows in the Eastern cluster. Specifically, how can increased precipitation potentially result in lower streamflow during summer? This seems counterintuitive and should be better explained or corrected.*

We argued that increases in winter precipitation ($P_{DJF}$) might have indirectly contributed to decreases in summer low flows in the eastern cluster via feedbacks with evapotranspiration ($E$). Specifically, increases in $P_{DJF}$ might have supported increases in storage ($S$) recharge in winter and thus in $S$ levels at the beginning of the growing season. In turn, increases in $S$ levels at the beginning of the growing season potentially contributed to increased vegetation growth and therefore $E$, by further contributing to reduced summer low flows. While a rigorous attribution of increases in $E$ to their drivers would require additional data (e.g., on vegetation gorwth) and we acknowledge that this negative correlation may also be spurious, we rephrased L355–357 to make our line of reasoning clearer.

L355–357: Increases in winter storage recharge may also promote increases in vegetation growth and $E$, and thus decreases in summer low flows, as the negative correlation between trends in $P_{DJF}$ and in summer low flows in the eastern cluster may suggest (Fig. 7), even though this relationship may be spurious.

*2- Trend Analysis in Abstract and Main Text (L20–21):*

*The statement: "Summer low flows decreased (increased) significantly in 31% (2%) of the catchments, with a median trend of -3.7% per decade across all catchments." needs further examination. If 67% of the catchments showed no significant trend, what is the rationale for computing the median trend across all catchments, including those without significant trends? Additionally, it would be more informative to report the median trend separately for the 31% of catchments showing decreasing trends and the 2% with increasing trends. This would help to clarify the magnitude and direction of significant changes.*

We decided to compute trend statistics on all catchments, and not on catchments with significant trends only, because we believe that trend slopes can be informative of the general tendency to the increase or decrease even if statistically non-significant at a predefined level and for a specific trend test. In the case of summer low flows, trend slopes are negative for most catchments with a median trend of -3.7 % decade$^{-1}$ and an interquartile range of -7.5/-0.6 % decade$^{-1}$, as reported in the

manuscript, which points to a general tendency to the decrease of summer low flows in the study region between 1970 and 2019 (Fig. 4c, white boxplot). We acknowledge that this was not easy to grasp from the Abstract alone, which we rephrased as reported below. Furthermore, we agree that statistics on significant trends only may provide a more comprehensive picture of the observed changes in the region and therefore, we added them in a table in the supplement (Table 1 here, new Table S1 in the supplement), which we now refer to in the main text as reported below.

Table 1: Trends in summer low flows ($7dQ_{min, JJA}$) and their predictors (annual evapotranspiration, $E$, precipitation over summer, $P_{JJA}$, spring, $P_{MAM}$, and winter $P_{DJF}$) over 1970–2019.

| Variable | Catchments with positive trends [%] | Catchments with negative trends [%] | Median trend across all catchments [% decade$^{-1}$] | Catchments with significant positive trends [%] | Median trend across catchments with significant positive trends [% decade$^{-1}$] | Catchments with significant negative trends [%] | Median trend across catchments with significant negative trends [% decade$^{-1}$] |
|---|---|---|---|---|---|---|---|
| $7dQ_{min, JJA}$ | 23 | 77 | -3.7 | 2 | 7.4 | 31 | -9.7 |
| $E$ | 66 | 34 | 1.1 | 27 | 4.6 | 5 | -4 |
| $P_{JJA}$ | 52 | 48 | 0.2 | 0 | - | 0 | - |
| $P_{MAM}$ | 34 | 66 | -0.6 | 1 | 4.8 | 3 | -6.9 |
| $P_{DJF}$ | 90 | 10 | 3.6 | 10 | 6.9 | 0 | - |

Abstract, L20–22: Summer low flows generally showed a decreasing tendency (median trend of -3.7 % decade$^{-1}$ and interquartile range of -7.5/-0.6 % decade$^{-1}$ across all catchments), significant negative trends in 31 % of the catchments, and significant positive trends in 2 % of them only.

L229–232: Trends in $7dQ_{min, JJA}$ were significantly negative in 31 % of the catchments and positive in 2 % of them only (Fig. 4b, Table S1). Across all catchments, trends in summer low flows had a median (interquartile range, IQR) of -3.7 (-7.5/-0.6) % decade$^{-1}$ (Fig. 4c). Focusing on catchments with significant negative trends only, a median decrease of -9.7 % decade$^{-1}$ was observed (Table S1).

L318–320: Summer low flows showed a median trend of -3.7 % decade$^{-1}$ (IQR of -7.5/-0.6 % decade$^{-1}$) across all catchments, with significant negative trends in 31 % and significant positive trends in 2 % of the catchments (Fig. 4b and c).

*3- Ambiguity in Trend Percentages (L225–227):*

*The sentence: "Trends in 7dQmin, JJA were significantly negative in 31% of the catchments and significantly positive in 2% of them (negative in 77% and positive in 23%, Fig. 4b), with a median (interquartile range, IQR) of -3.7 (-7.5/-0.6)% per decade across all catchments (Fig. 4c)." is confusing. It is unclear how the 31% and 2% (which refer to statistically significant trends) relate to the 77% and 23% figures (which seem to include non-significant trends). Please clarify the distinction between statistically significant trends and overall direction of trends, and how these percentages are used in analysis and interpretation.*

Trends in summer low flows were negative in 77 % of the catchments and positive in the remaining 23 %. Moreover, trends were significantly negative in 31 % of the catchments and significantly positive in only 2 % of them. As better explained above, we used both the trend slopes and their significance for the interpretation of our results, since we believe they provide complementary information. While the magnitude of trend slopes across all catchments provides an indication of the general tendency to the decrease/increase, the significance of these slopes provides information regarding the strength of these changes. We see, however, that reporting both percentages in the same sentence may hamper

its readability and thus, we decided to move the percentages of positive and negative trends in the new Table S1 (see reply to comment #2).

L229–230: Trends in 7d$Q_{min, JJA}$ were significantly negative in 31 % of the catchments and positive in 2 % of them only (Fig. 4b, Table S1).

*4- Over-Reliance on Previous Studies:*

*While referencing previous literature is important for contextualization, several of the author's responses rely heavily on prior studies without providing sufficient evidence from the current analysis. It is recommended to strengthen the manuscript by presenting clearer links between your results and your arguments, supported by direct evidence from your own data.*

We would like to point out that we performed a number of new analyses of our dataset for our first round of responses, rather than mainly referring to previous studies as this comment seems to suggest to us. Specifically, we added (i) the catchment-scale attribution of trends in summer low flows through multiple linear regression for each catchment (Referee #1, Comment #3), (ii) the analysis of trends in annual streamflow and its timing (Referee #2, Comment #1), and (iii) a formal verification of the applicability of the method we used to estimate the dynamic storage of the catchments (Referee #2, Comment #2). For all these analyses, we introduced methodological details and subsequent results in the manuscript or the supplementary material. By carefully going through our previous replies, we furthermore noticed that we referred to previous studies in the replies to:

1. Referee #1, Comment #1, to support our assumption of little influence of the *P* correction procedure on our results;
2. Referee #1, Comment #2 and Referee #2, Comment #5, to support our choice of using *P* in the preceding seasons as proxy for *S* conditions;
3. Referee #2, Comment #3 and Referee #2, Comment #7, to refer to previous works which already investigated what suggested by the Reviewer;
4. Referee #2, Comment #4, to support our choice of using the water-balance approach for *E*.

We do not list here replies where we used previous studies as references for our methods (e.g., Referee #2, Comment #1) or as additional evidence for our findings/argumentations (e.g., Referee #1, Comment #4 and Referee #2, Comment #1), since we believe this is not what you are pointing to here. Among the above-listed instances, we see only the first one as an assumption, based on previous findings rather than new data analyses, whereas we see the others as appropriate references to works using similar methods (points 2 and 4) or investigating partly-related topics (point 3).

Therefore, we have now formally tested our initial speculation regarding point 1. Specifically, we repeated all our analyses with uncorrected *P* for gauge undercatch. Using uncorrected *P* results in a median trend in *P* equal to 0.4 % decade$^{-1}$ and in *E* equal to 1.2 % decade$^{-1}$ across all catchments (Fig. 1 here), as compared to 0.3 % decade$^{-1}$ and 1.1 % decade$^{-1}$ respectively when using corrected *P*. Importantly, these differences in trend magnitudes do not affect the attribution of trends in summer low flows to their main predictors, with summer *P* ($P_{JJA}$) still the dominant predictor for the pre-Alpine cluster and *E* as a significant predictor for the eastern and northern clusters, with $P_{JJA}$ in the latter case, according to both the analyses that we performed to this end (Fig. 2 and 3 here). Also the results regarding the changes in the *P-Q* relationship during the multi-year drought between 1989 and 1993 still hold when using uncorrected *P*, with the multi-year drought detected for 14 % of catchments and changes in 33 % of these, as compared to the 15 % and 26 % obtained with corrected *P*, and a median trend in *E* for catchments with changes equal to 5.9 % decade$^{-1}$, as compared to the 6.1 % decade$^{-1}$ for corrected *P* (Fig. 4 here). Thus, our conclusions do not depend on the use of the correction procedure. We added this analysis in the revised version of the manuscript, by introducing it in the Methods,

Results, and Discussion as follows, and reporting Fig. 1 and 4 here as new Fig. S6 and S10 in the supplement.

[Figure]

Fig. 1: Long-term variations in predictors of variations in summer low flows over 1970–2019 (annual evapotranspiration, *E*, panels a–c, precipitation over summer, $P_{JJA}$, panels d–f, spring, $P_{MAM}$, panels g–i, and winter $P_{DJF}$ panels j–l) from precipitation not corrected for gauge undercatch (Sect. 2.2). (a, d, g, and j) Average anomalies across the catchments. (b, e, h, and k) Maps of catchment-scale trends (black edges if significant). (c, f, i, and l) Boxplots of trends for all catchments and by cluster.

[Figure]

Fig. 2: Attribution of long-term variations in summer low flows to their predictors (contribution to temporal dynamics) from precipitation not corrected for gauge undercatch (Sect. 2.2): relative contribution of annual evapotranspiration (*E*), summer

($P_{JJA}$), spring ($P_{MAM}$), and winter precipitation ($P_{DJF}$) to the predicted long-term dynamics of summer low flows ($7dQ_{min, JJA}$) from multiple linear regression (Sect. 2.5) for the different clusters. Pre-al. refers to pre-Alpine, South. to south-central, East. to eastern, and North. to northern cluster. Vertical lines indicate the uncertainty of the regression coefficients.

[Figure]

Fig. 3: Attribution of long-term variations in summer low flows to their predictors (strength of spatial coherence) from precipitation not corrected for gauge undercatch (Sect. 2.2): Pearson's correlation coefficients ($r$) between catchment-scale trends in summer low flows ($7dQ_{min, JJA}$) and in potential predictors (annual evapotranspiration, $E$, summer precipitation $P_{JJA}$, spring precipitation $P_{MAM}$, and winter precipitation $P_{DJF}$) over 1970–2019, for the catchments in the different clusters. Pre-al. refers to pre-Alpine, South. to south-central, East. to eastern and North. to northern cluster.

[Figure]

Fig. 4: Changes in the annual relationship between precipitation ($P$) and streamflow ($Q$, $P$-$Q$ relationship) during the multi-year drought between 1989 and 1993, and their potential predictors, from $P$ not corrected for gauge undercatch (Sect. 2.2). (a) Map of the magnitude of changes in the $P$-$Q$ relationship ($C_{P-Q\,rel}$) across the study catchments. (b) Histogram of $C_{P-Q\,rel}$. (c) Boxplots of trends in annual catchment actual evapotranspiration ($E$) and $P$ over 1970–1993, for catchments with change and no change in the $P$-$Q$ relationship. (d) Boxplots of mean anomalies in $E$ and $P$ over the drought, for catchments with change and no change in the $P$-$Q$ relationship.

Methods, L145–147: We then computed area-weighted catchment average time series from $P_{corr}$ ($P$ in the following) and $P_{uncorr}$ fields for each catchment. We repeated all analyses (Fig. 1) for both $P$ and $P_{uncorr}$ to verify that the use of the correction procedure does not affect out results.

Results, L255–256: Similar trends, for both $E$ and seasonal $P$, were obtained when using $P_{uncorr}$ (Fig. S6).

L295–296: When using alternative low-flow metrics ($30dQ_{min, JJA}$ and $7dQ_{min, M-O}$) instead of $7dQ_{min, JJA}$ or $P_{uncorr}$ instead of $P$, results were comparable to those presented here for both attribution analyses (not shown).

L308–309: The use of $P_{uncorr}$ led to comparable results (Fig. S10).

Discussion, L434–435: This correction procedure may lead to further uncertainties, but we achieved comparable results for all analyses for both *P* and $P_{uncorr}$ (Sect. 3.2 and 3.3), meaning that our conclusions do not depend on the use of the correction procedure.

*5- Language and Grammar Corrections:*

*Please revise the manuscript for language accuracy. For example, in Section 4.4: "Due to the generally coarse resolution of satellite-derived E estimates, we computed E from observed P and Q, and estimates of the Sdyn of the catchments from Q data as a first order approximation of S." This sentence could be rewritten for clarity and correctness. A suggestion: "Given the generally coarse resolution of satellite-derived E estimates, we derived E using observed P, Q, and estimates catchment dynamic storage (Sdyn) from Q data, as a first-order approximation of S."*

Thank you for this valuable suggestion on how to improve the clarity of our writing. We had another careful round of proofreading for language polishing and we refer to the tracked-changes version of the manuscript for the edits we implemented to this end.

---

## Author Response (AR3)

Dear Editor,

We thank you for your time in reviewing our re-revised manuscript and your positive feedback on it. Please find below our reply to your comment (italic) and the changes we implemented in the manuscript to address it (underlined). We further attach a revised version of the manuscript. Finally, we would like to thank you and the Editorial Team for your collaboration throughout the whole peer review process.

Best regards,

Giulia Bruno and co-authors

*1-Clarification on $P_{DJF}$ and Summer Low Flows (Referee #1, Question 4):*

*Authors response: "We argued that increases in winter precipitation ($P_{DJF}$) might have indirectly contributed to decreases in summer low flows in the eastern cluster via feedbacks with evapotranspiration (E). Specifically, increases in $P_{DJF}$ might have supported increases in storage (S) recharge in winter and thus in S levels at the beginning of the growing season. In turn, increases in S levels at the beginning of the growing season potentially contributed to increased vegetation growth and therefore E, by further contributing to reduced summer low flows."*

*The conceptual linkage between increased precipitation, higher soil moisture or groundwater storage, enhanced vegetation, increased evapotranspiration is well-documented in ecohydrology and hydrological literature. However, in your response, the term storage (S) is ambiguous. Please clarify whether you are referring to soil moisture storage or groundwater storage. While I agree that increased vegetation growth can lead to higher evapotranspiration and a subsequent reduction in soil moisture, this mechanism does not directly apply to groundwater storage. On the contrary, an increase in winter precipitation that enhances groundwater recharge would typically support higher summer low flows, not lower. This appears to contradict the mechanism you propose. Additionally, I encourage you to address the directionality of this relationship. If a decrease in winter precipitation ($P_{DJF}$) will also lead to a reduction in summer low flows, how do you reconcile that with the claim that an increase in $P_{DJF}$ also results in reduced summer low flows? If both increasing and decreasing $P_{DJF}$ will lead to the same outcome (reduced summer low flows), a clearer justification or an alternative explanation is needed to resolve this apparent inconsistency.*

We understand that two pieces of clarification are required: (i) whether we referred to soil moisture or groundwater storage in the previous reply, and (ii) on the directionality of the link between winter precipitation ($P_{DJF}$) and summer low flows.

Regarding the first point, we would like to highlight that we do not rely on soil moisture or groundwater data, due to their unavailability across all the study catchments. For the trend attribution, we approximated sub-surface storage conditions at the beginning of summer with precipitation in the preceding seasons. This approach does not allow us to disentangle the

contribution of the two storages (soil moisture and groundwater) and therefore, we used the overall term 'storage' ($S$) in our previous reply and in the manuscript. To avoid confusion, we added this point at L53–55 when we first introduce the term 'storage' and its abbreviation.

Long-term decreases in summer low flows in small catchments may originate from long-term increases in catchment actual evapotranspiration ($E$), decreases in precipitation ($P$) during the summer season, and decreases in water storage in the catchments ($S$, mainly in the soil and groundwater in catchments with little influence of snow, Montanari et al., 2023).

Regarding the second point on the link between $P_{DJF}$ and summer low flows, we argue that increases in $P_{DJF}$ may lead to increases in soil moisture which may on the one hand translate into increased groundwater storage and summer low flows. On the other hand, increases in $P_{DJF}$ and soil moisture storage may also lead to increased vegetation growth and $E$ during the growing season. If the latter process dominates, increases in $P_{DJF}$ could result in decreases in soil moisture storage and groundwater recharge during the growing season, which may ultimately support decreases in summer low flows, as the negative correlation between trends in $P_{DJF}$ and in summer low flows in the eastern cluster may suggest, even though we acknowledge that this relationship may also be spurious. To make our line of reasoning clearer, we expanded L355–359 as follows.

From a mechanistic point of view, increases in winter storage recharge may also promote increases in vegetation growth and $E$, and thus decreases in soil moisture storage and groundwater recharge during the growing season and ultimately in summer low flows. We speculate here that the negative correlation between trends in $P_{DJF}$ and in summer low flows in the eastern cluster may suggest this chain of processes (Fig. 7), even though we acknowledge that this relationship may be spurious.